# Adaptive Debiasing Tsallis Entropy for Test-Time Adaptation

**Xiangyu Wu**[*][♠][◇] **Dongming Jiang**[*][♣] **Feng Yu**[*][♠] **Yueying Tian**[♡] **Jiaqi Tang**[△]
**Qing-Guo Chen**[◇] **Yang Yang**[†][♠] **Jianfeng Lu**[†][♠]
[♠]Nanjing University of Science and Technology [♣]University of Texas at Dallas
[◇]Alibaba Cloud [♡]University of Sussex [△]Hong Kong University of Science and Technology

## Abstract

Mainstream Test-Time Adaptation (TTA) methods for adapting vision-language models, *e.g.*, CLIP, typically rely on Shannon Entropy (`SE`) at test time to measure prediction uncertainty and inconsistency. However, since CLIP has a built-in bias from pretraining on highly imbalanced web-crawled data, `SE` inevitably results in producing biased estimates of uncertainty entropy. To address this issue, we notably find and demonstrate that Tsallis Entropy (`TE`), a generalized form of `SE`, is naturally suited for characterizing biased distributions by introducing a non-extensive parameter $q$, with the performance of `SE` serving as a lower bound for `TE`. Building upon this, we generalize `TE` into **A**daptive **D**ebiasing **T**sallis **E**ntropy (`ADTE`) for TTA, customizing a class-specific parameter $q^l$ derived by normalizing the estimated label bias from continuously incoming test instances, for each category. This adaptive approach allows `ADTE` to accurately select high-confidence views and seamlessly integrate with label adjustment strategy to enhance adaptation, without introducing distribution-specific hyperparameter tuning. Besides, our investigation reveals that both `TE` and `ADTE` can serve as direct, advanced alternatives to `SE` in TTA, without any other modifications. Experimental results show that `ADTE` outperforms state-of-the-art methods on ImageNet and its five variants, and achieves the highest average performance on 10 cross-domain benchmarks, regardless of the model architecture or text prompts used. Our code is available at https://github.com/Jinx630/ADTE.

## 1 Introduction

Vision-Language Models (VLMs) (Radford et al., 2021; Wu et al., 2022; Zeng et al., 2024; Wu et al., 2025d; Li et al., 2025; Jiang et al., 2025; Yu et al., 2025; Zeng et al., 2025b), pretrained on large-scale datasets (Sharma et al., 2018; Schuhmann et al., 2022), exhibit remarkable generalization abilities across various downstream tasks. Despite this, they are susceptible to performance degradation when confronted with considerable discrepancies between training and testing domains. To mitigate this, a technology known as Test-Time Adaptation (TTA) (Shu et al., 2022; Zhu et al., 2024; Zhou et al., 2025; Wu et al., 2025c), enables models to adapt instantaneously to diverse instance distributions during testing, in contrast to earlier prompt learning techniques (Zhou et al., 2022; Hu et al., 2023; Xing et al., 2024; Wu et al., 2024; Huang et al., 2024) that require complex training procedures.

Among these representative TTA works, TPT (Shu et al., 2022) and its enhancement, DiffTPT (Feng et al., 2023), learn instance-level prompts by selecting high-confidence augmented views for each test instance. Zero (Farina et al., 2024) simplifies the TTA process, demonstrating that the prediction of the marginal probability distribution remains approximately invariant under entropy minimization. ML-TTA (Wu et al., 2025c), equipped with Bound Entropy Minimization (`BEM`), enables the adaptation of multi-label instances. The central idea is to select lower entropy views as the high-confidence views, aiming to reduce uncertainty among these views, a readily demonstrable theory.

---

[*]Equal contribution.
[†]Corresponding author.

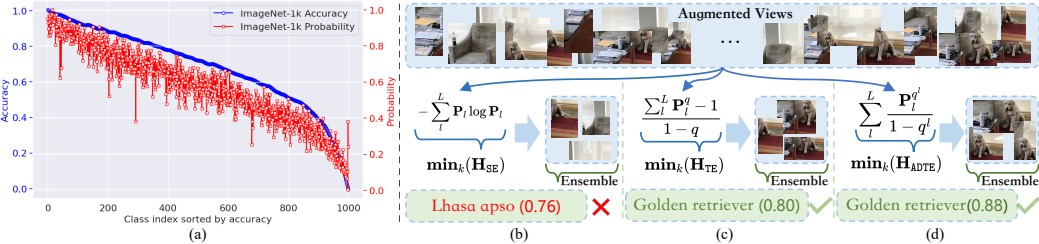

Figure 1: (a) VLM bias, showing higher confidence and accuracy for `head` classes and lower confidence and accuracy for `tail` classes. (b) The standard Shannon Entropy (`SE`)-based method is widely used in TTA. (c) and (d) Our proposed method, which uses Tsallis Entropy (`TE`) and Adaptive Debiasing Tsallis Entropy (`ADTE`) for selecting high-confidence views.

However, VLMs (*e.g.*, CLIP), as discussed in research (Allingham et al., 2023; Parashar et al., 2024; Zhu et al., 2023; 2024; Wu et al., 2025a; Ke et al., 2025), are pretrained on imbalanced web-scale datasets, inevitably possessing inherent prediction bias. This bias causes the model to consistently exhibit `low/high` confidence in the `tail/head` categories, leading to `lower/higher` accuracy, as shown in Figure 1 (a). For this reason, during the TTA process, the predicted probability for a category $l$, denoted as $\mathbf{p}$, may significantly differ from its true unbiased probability $\hat{\mathbf{p}}$ (*i.e.*, $|\mathbf{p} - \hat{\mathbf{p}}| > 0$). Methods based on Shannon Entropy (`SE`), defined as $\mathbf{H}_{\text{SE}} = -\sum_l \mathbf{p} \log \mathbf{p}$, are impacted because their entropy calculations rely on these potentially biased probabilities $\mathbf{p}$. Moreover, from its definition, `SE` fails to account for varying degrees of bias in probabilities across different classes (*i.e.,* `head`, `middle`, and `tail` classes). Instead, `SE` applies a uniform computation formula ($-\mathbf{p} \log \mathbf{p}$) across all probabilities, as shown in Figure 1 (b). As a result, in TTA methods that rely on `SE`, the entropy values estimated for each augmented view are themselves biased. This bias, in turn, affects the selection of high-confidence views from a set of augmented views.

Our first key observation is that Tsallis Entropy (`TE`) (Tsallis, 1988; 1998), a generalization of `SE`, is well-suited for characterizing uncertainty in the presence of biased distributions. By introducing an additional parameter $q$[1], `TE` can characterize likelihoods exhibiting statistical dependence and effectively mitigate the impact of bias. First, we demonstrate that `TE` is a limiting case of `SE` when $q \to 1$. On the other hand, when $q < 1$, `TE` tends to select more confident views (characterized by higher $\mathbf{Tcr}_K$ in Definition 1), implying that the performance of `SE` can be regarded as a lower bound for that of `TE`. In other words, there must exist an appropriate parameter $\hat{q}$ such that models using `TE` outperform those using `SE` in the selection of high-confidence views. Subsequently, we demonstrate that when $q < 1$, `TE` can effectively alleviate the impact of inherent bias for VLMs.

However, there are two major difficulties in applying `TE` in TTA practice: **(i)** manually tuning the optimal value of $q$ in standard `TE` is impractical for various test distributions; and **(ii)** the optimal parameter $\hat{q}$ for each category may also differ, since each category is affected by bias differently, as discussed earlier. Therefore, we generalize `TE` into **A**daptive **D**ebiasing **T**sallis **E**ntropy (`ADTE`) for TTA, customizing a class-specific $q^l$ for category $l$, as shown in Figure 1 (d). These parameters are derived via min–max normalization of the estimated label biases from continuously incoming test instances. Significantly, both `TE` and `ADTE` can serve as direct, advanced alternatives to `SE` and integrate seamlessly with the logit adjustment strategy to enhance adaptation performance. Experiments show that, irrespective of the model architecture or text prompts employed, `ADTE` surpasses the SOTAs on ImageNet and its five variants, achieving the highest average performance across 10 cross-domain benchmarks.

## 2 RELATED WORKS

**Test-Time Adaptation (TTA)**. TTA methods (Shu et al., 2022; Karmanov et al., 2024; Zhu et al., 2024; Wu et al., 2025c; Zhou et al., 2025) dynamically adjust pre-trained models using unlabeled test data during inference, *e.g.*, detection (Ouyang et al., 2024; Zhao, 2024; Edstedt et al., 2024; Zhao et al., 2026) tasks. TTA has been explored in various settings, including "fully" TTA (Wang et al., 2021; Zhao et al., 2023a), "online" TTA (Lee & Chang, 2024; Lee et al., 2024a), and "continuous" TTA (Liu et al., 2024; Song et al., 2023). TPT (Shu et al., 2022) is one of the first works

---

[1]$q$ is called *non-extensive parameter* in the literature (Tsallis, 1988; 1998).

to apply prompt tuning for adapting VLMs to previously unseen distributions. Zero (Farina et al., 2024) demonstrates that minimizing entropy does not alter the dominant class prediction, providing a theoretical basis for this approach. Other works have introduced new techniques: Frolic (Zhu et al., 2024) uses label-free prompt distribution learning and bias correction, ML-TTA (Wu et al., 2025c) employs a bound entropy minimization objective in multi-label scenarios, and BCA (Zhou et al., 2025) leverages Bayesian principles to refine predictions as new data arrives.

**Entropy-based Uncertainty Minimization in Adaptation.** Entropy Minimization (EM) (Grandvalet & Bengio, 2004; Berthelot et al., 2019; Gilo et al., 2024; Yang et al., 2024; Wan et al., 2024; Wu et al., 2025b; Dai et al., 2025; Zeng et al., 2025a) is a common strategy for reducing prediction uncertainty and promoting clearer decision boundaries. Early work (Grandvalet & Bengio, 2004) used minimum entropy regularization to leverage unlabeled data, while MME (Saito et al., 2019) and Tent (Wang et al., 2021) showed its effectiveness for adapting to varied test distributions. Recently, Tsallis Entropy (TE) has emerged as an alternative with a tunable parameter $q$ controlling the sharpness of the entropy landscape. In domain adaptation and self-training (Lu et al., 2023; Liu et al., 2021; Lee et al.; Zhao et al., 2023b), TE has been optimized to improve pseudo-label reliability and robustness to noise in supervised or source-free settings, focusing on feature representation learning. While EM minimizes standard Shannon entropy, TE offers a more flexible formulation that may extend to online test-time adaptation.

**Logit Adjustment (LA).** LA (Menon et al., 2021; Li et al., 2022; Zhao et al., 2024; Xu et al., 2024; Jia et al., 2024; Miao et al., 2024) is primarily used for long-tailed recognition and class-imbalanced learning. It adjusts a model's logits to compensate for biases caused by imbalanced training data. GCL (Li et al., 2022) adds a Gaussian perturbation to logits, giving larger perturbations to `tail` classes to enhance their gradient contribution. LoTNext (Xu et al., 2024) introduces graph-based and long-tailed loss adjustments to improve spatial and temporal prediction. HTC (Jia et al., 2024) addresses the long-tail issue with candidate label set disambiguation, class distribution estimation, and classifier weight estimation. COCL (Miao et al., 2024) combines debiased large-margin learning and outlier-class-aware logit calibration to effectively mitigate biases. We employ the logit adjustment to estimate the class-specific parameter $q^l$ for each category in Tsallis Entropy.

## 3 PRELIMINARIES

Consider the CLIP (Radford et al., 2021) model pretrained on $\mathcal{D}^{\text{train}} \coloneqq \{(\mathbf{x}_i^{\text{train}}, \mathbf{y}_i^{\text{train}}) \mid \mathbf{x}_i^{\text{train}} \in \mathcal{X}^{\text{train}}, \mathbf{y}_i^{\text{train}} \in \mathcal{Y}^{\text{train}}\}_{i=1}^{M^{\text{train}}}$ and a downstream test set $\mathcal{D}^{\text{test}} \coloneqq \{(\mathbf{x}_i^{\text{test}}, \mathbf{y}_i^{\text{test}}) \mid \mathbf{x}_i^{\text{test}} \in \mathcal{X}^{\text{test}}, \mathbf{y}_i^{\text{test}} \in \mathcal{Y}^{\text{test}}\}_{i=1}^{M^{\text{test}}}$, which may follow an arbitrary distribution. We illustrate the standard Test-Time Adaptation (TTA) process with Zero (Farina et al., 2024), which consists of *Random view augmentation*, *Confident views selection*, and *Confident views ensemble*.

***Random View Augmentation***. Given a test instance $\mathbf{x}^{\text{test}}$ from $\mathcal{D}^{\text{test}}$ and a set $\mathcal{A}$ of $N$ random augmentation functions, $\mathbf{x}^{\text{test}}$ is augmented $N$ times to create a set of diverse views, which are denoted as $\mathbf{X}^{\text{test}} \coloneqq \{\mathbf{x}_j^{\text{test}} \mid \mathbf{x}_j^{\text{test}} = \mathcal{A}_j(\mathbf{x}^{\text{test}})\}_{j=1}^{N}$.

***Confident Views Selection***. In information theory, Shannon Entropy (SE, (Shannon, 1948)) is commonly used to quantify uncertainty[2]. For each view, uncertainty is measured by SE, defined as:

$$\mathbf{H}_{\text{SE}}(\mathbf{P}(\cdot \mid \mathbf{x}_j^{\text{test}})) = -\sum_{l=1}^{L} \mathbf{P}(y = l \mid \mathbf{x}_j^{\text{test}}) \log[\mathbf{P}(y = l \mid \mathbf{x}_j^{\text{test}})], \quad (1)$$

where $l \in \mathcal{Y}^{\text{test}}$ denotes a class label, and $L$ is the number of classes in the test set. The core of Zero (Farina et al., 2024) lies in minimizing the marginal entropy over the prediction distributions corresponding to the selected high-confidence augmented views (*i.e.,* views with lower entropy) by a ratio $\tau$, which reduces the model's uncertainty and prediction inconsistency across these views. However, due to the inherent prediction bias of models like CLIP, the estimated SE values for `head` or `tail` categories are themselves biased. This bias in SE estimation, in turn, affects the selection of high-confidence views.

---

[2]Thermodynamic entropy, an earlier concept, is typically expressed via Boltzmann-Gibbs-Shannon entropy $\mathbf{H}_{\text{BGS}} = -k \sum_i p_i \log p_i$, where $k$ is the Boltzmann constant. $\mathbf{H}_{\text{BGS}}$ is mathematically identical to Shannon Entropy, but generally less familiar to the machine learning community

***Confident Views Ensemble***. Unlike TPT (Shu et al., 2022), which uses high-confidence views to update the model's prompts during TTA, Zero (Farina et al., 2024) simplifies the TTA process. It argues that the updated model's prediction remains invariant under entropy minimization, which allows the final prediction to be obtained by directly aggregating the high-confidence views. The model then immediately adapts to the next test instance. Due to its simplicity and effectiveness, SE has become a standard metric for high-confidence view selection in recent TTA methods.

## 4 METHOD

### 4.1 TSALLIS ENTROPY

When multiple probability distributions are involved, Shannon entropy $\mathbf{H}_{\text{SE}}$ quantifies total uncertainty under the *extensivity* assumption, meaning that the entropy of independent parts simply adds, i.e., $\mathbf{H}_{\text{SE}}(\{A, B\}) = \mathbf{H}_{\text{SE}}(A) + \mathbf{H}_{\text{SE}}(B)$. In the real world, such as with biased model predictions, this additive property does not hold. To capture such *non-extensivity*, TE generalizes SE by introducing a parameter $q$ that characterizes the degree of non-additivity, defined as:

$$\mathbf{H}_{\text{TE}}(\mathbf{P}(\cdot \mid \mathbf{x}_j^{\text{test}})) = \frac{1}{1-q}\left(\sum_{l=1}^{L}\mathbf{P}(y = l \mid \mathbf{x}_j^{\text{test}})^q - 1\right), \qquad (2)$$

where $q \in \mathbb{R}$ is the additional hyperparameter. Using TE, the total entropy can be calculated as $\mathbf{H}_{\text{TE}}(\{A, B\}) = \mathbf{H}_{\text{TE}}(A) + \mathbf{H}_{\text{TE}}(B) + (1 - q)\mathbf{H}_{\text{TE}}(A)\mathbf{H}_{\text{TE}}(B)$, where the additional term $(1 - q)\mathbf{H}_{\text{TE}}(A)\mathbf{H}_{\text{TE}}(B)$ is used to characterize influence between components. We first demonstrate that TE possesses the following crucial properties, which make SE the lower bound in performance for TE. In other words, there exists an appropriate parameter $\hat{q}$ for which models based on TE are capable of selecting more accurate and confident views compared to those relying on SE.

▶ `Top-K Cumulative Reliability:`

**Definition 1.** *VLMs like CLIP classify images by computing similarity scores between an image and "a photo of {classes}". For an augmented view $\mathbf{x}_j^{\text{test}}$, the similarity with a class $l$ is $\mathbf{z}_l^{\top}\mathbf{x}_j^{\text{test}}$, where $\mathbf{z}_l^{\top}$ denotes the textual embedding representing class $l$. We denote the Top-K Cumulative Reliability $\mathbf{Tcr}_K$ as the sum of the $K$ highest similarity scores. Mathematically, it is given by:*

$$\mathbf{Tcr}_K(\mathbf{x}_j^{\text{test}}) = \sum_{l \in \mathcal{T}_K}\mathbf{z}_l^{\top}\mathbf{x}_j^{\text{test}}, \qquad (3)$$

*where $\mathcal{T}_K$ is the set of indices corresponding to the $K$ highest similarity scores. Actually, CLIP typically achieves an extremely high Top-K (e.g., K=5 or K=10) accuracy, indicating that although CLIP may not always predict the exact label, it is proficient at generating a candidate set that contains the correct label.*

▶ `Shannon-Tsallis` $q \to 1$ `Equivalence:`

**Property 1.** *As $q \to 1$, TE becomes equivalent to SE:*

$$\lim_{q \to 1}\mathbf{H}_{\text{TE}}(\mathbf{P}(\cdot \mid \mathbf{x}_j^{\text{test}})) = \mathbf{H}_{\text{SE}}(\mathbf{P}(\cdot \mid \mathbf{x}_j^{\text{test}})). \qquad (4)$$

*See the Appendix B.1 for the detailed proof.* Property 1 demonstrates that TE is a generalization of SE, as illustrated in Figure 2 for a case with two classes. This property confirms that TE is consistent with the well-established SE theory and can be used to analyze more intricate non-extensive probability distributions by adjusting $q$.

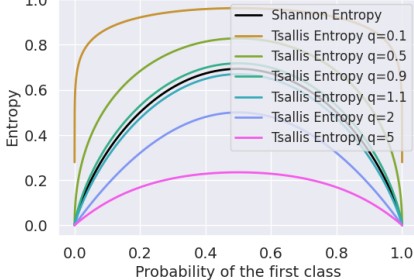

Figure 2: Comparison between SE and TE.

▶ `Higher` $\mathbf{Tcr}_K$ `under TE as` $q \searrow$ `and Comparison with SE:`

**Property 2.** *Through experimental analysis, we find that as the parameter $q$ decreases, the set of high-confidence views selected by TE tends to have a higher average $\mathbf{Tcr}_K$ value (for $K > 1$). For two different parameter values $q_1$ and $q_2$ with $q_1 < q_2$, and their corresponding selected view sets $\mathcal{X}_{q_1}$ and $\mathcal{X}_{q_2}$ of equal size, we observe:*

$$\frac{1}{|\mathcal{X}_{q_1}|}\sum_{\mathbf{x_1} \in \mathcal{X}_{q_1}}\mathbf{Tcr}_K(\mathbf{x_1}) > \frac{1}{|\mathcal{X}_{q_2}|}\sum_{\mathbf{x_2} \in \mathcal{X}_{q_2}}\mathbf{Tcr}_K(\mathbf{x_2}). \qquad (5)$$

*Furthermore, there exists a particular $q^*$ and corresponding view set $\mathcal{X}_{q^*}^{\text{TE}}$, TE outperforms SE in this regard, with the view set selected by SE denoted as $\mathcal{X}^{\text{SE}}$:*

$$\frac{1}{|\mathcal{X}_{q^*}^{\text{TE}}|} \sum_{\mathbf{x_1} \in \mathcal{X}_{q^*}^{\text{TE}}} \mathbf{Tcr}_K(\mathbf{x_1}) > \frac{1}{|\mathcal{X}^{\text{SE}}|} \sum_{\mathbf{x_2} \in \mathcal{X}^{\text{SE}}} \mathbf{Tcr}_K(\mathbf{x_2}). \tag{6}$$

*See the Appendix B.2 for the detailed experimental analysis.* Property 2 indicates that, in contrast to SE, TE tends to select views with higher $\mathbf{Tcr}_K$, and the corresponding ground truth has a higher similarity level. The core objective of TTA is to select the most confident and accurate views.

## 4.2 CORRECTING BIASED ENTROPY EFFECT WITH TE

With the theoretical properties of TE established, we now investigate how its additional parameter $q$ can mitigate the biased entropy effect observed with SE in VLMs. For a tail category, the biased prediction $\mathbf{p}$ and unbiased prediction $\hat{\mathbf{p}}$ are both close to 0, with $\hat{\mathbf{p}} > \mathbf{p}$ (based on empirical results; this is likely due to the CLIP model being very uncertain about tail predictions, as shown in Figure 1 (a)). This results in a biased entropy value, where $(-\hat{\mathbf{p}} \log \hat{\mathbf{p}}) - (-\mathbf{p} \log \mathbf{p}) > 0$.

To analyze the effect of TE, we rewrite Equation 2 to examine the entropy value of each category:

$$\mathbf{H}_{\text{TE}}(\mathrm{P}) = \frac{\sum_{l=1}^{L} \mathrm{P}_l^q - 1}{1 - q} = \sum_{l=1}^{L} \frac{\mathrm{P}_l^q}{1 - q} - \frac{1}{1 - q} = \sum_{l=1}^{L} \frac{\mathrm{P}_l^q}{1 - q} - \mathbf{C}, \tag{7}$$

where $\mathrm{P}_l^q := \mathbf{P}(y = l \mid \mathbf{x}_j^{\text{test}})^q$, and $\mathbf{C}$ is a constant. Since the constant $\mathbf{C}$ does not affect the ranking of entropy values, we can focus only on the term $\frac{\mathbf{p}^q}{1-q}$. For TE to correct the bias of SE, we require the inequality $\mathbf{F}(\mathbf{p}, q) = \frac{\mathbf{p}^q}{1-q} - (-\mathbf{p} \log \mathbf{p}) > 0$ to hold. The numerical value of $\mathbf{F}(\mathbf{p}, q)$ reflects the degree to which TE corrects the bias entropy of SE. We analyze the behavior of this function for a tail category probability $\mathbf{p} \in (0, \epsilon)$, where $\epsilon > 0$ is a small value:

▶ (1).As $q \to +\infty$, we have $\lim_{\mathbf{p} \to 0^+} \mathbf{F}(\mathbf{p}, q) = 0^-$ and $\forall q_1 < q_2 < +\infty, \Rightarrow \mathbf{F}(\mathbf{p}, q_1) < \mathbf{F}(\mathbf{p}, q_2) < 0$.

▶ (2).As $q \to 1^+$, we have $\lim_{\mathbf{p} \to 0^+} \mathbf{F}(\mathbf{p}, q) = -\infty$ and $\forall 1 < q_1 < q_2, \Rightarrow \mathbf{F}(\mathbf{p}, q_1) < \mathbf{F}(\mathbf{p}, q_2) < 0$.

▶ (3).As $q \to 0^+$, we have $\lim_{\mathbf{p} \to 0^+} \mathbf{F}(\mathbf{p}, q) = 1^-$ and $\forall 0 < q_1 < q_2, \Rightarrow \mathbf{F}(\mathbf{p}, q_1) > \mathbf{F}(\mathbf{p}, q_2) > 0$.

▶ (4).As $q \to 1^-$, we have $\lim_{\mathbf{p} \to 0^+} \mathbf{F}(\mathbf{p}, q) = +\infty$ and $\forall q_1 < q_2 < 1, \Rightarrow \mathbf{F}(\mathbf{p}, q_2) > \mathbf{F}(\mathbf{p}, q_1) > 0$.

*See the Appendix B.3 for the detailed proof.* Based on Property 1 (when $q \to 1$, TE and SE become equivalent), conclusions (2) and (4) are not applicable. Conclusion (1) also does not meet our requirements since $\mathbf{F}(\mathbf{p}, q) < 0$. Conclusion (3) shows that when $0 < q < 1$, TE can naturally mitigate the effect of VLM bias, with the correction magnitude increasing as $q$ decreases.

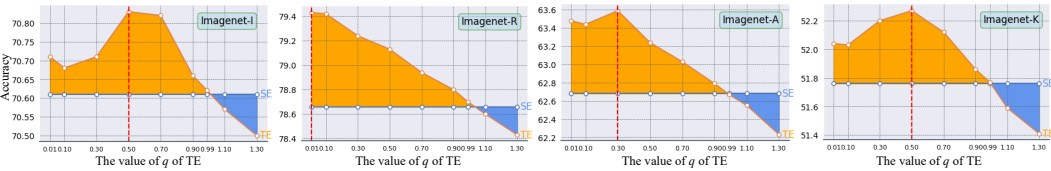

Figure 3: TE at different $q$ values vs. SE; the red dashed line marks the optimal $q$ of TE.

## 4.3 ADAPTIVE DEBIASING TSALLIS ENTROPY (ADTE)

A limitation of the standard TE formulation is that $q$ is a manually-tuned hyperparameter. A value of $q > 1$ can exacerbate the bias, while a value too close to 0 can overcorrect it. As shown in Figure 3, the optimal $q$ varies across different test distributions, making manual tuning impractical in a streaming environment. To address this, we generalize TE into **A**daptive **D**ebiasing **T**sallis **E**ntropy (ADTE), which customizes a class-specific parameter $q^l$ for each category $l$, allowing the model to adapt to any test distribution. Building upon Equation 7, ADTE is defined as:

$$\mathbf{H}_{\text{ADTE}}(\mathrm{P}) = \sum_{l=1}^{L} \frac{\mathrm{P}_l^{q^l}}{1 - q^l}. \tag{8}$$

---

**Algorithm 1** Estimating the bias $\mathbf{b}$

1: **Input**: Estimated $\mathbb{E}_{\mathbf{x}\sim\mathbf{P}(\mathbf{x}|l')}[\mathbf{P}(l|\mathbf{x})]$, maximum number of iteration $T$, convergence threshold $\varepsilon$
2: **Output**: Estimated $\mathbf{b}$
3: $a_{ll'} := \mathbb{E}_{\mathbf{x}\sim\mathbf{P}(\mathbf{x}|l')}[\mathbf{P}(l|\mathbf{x})]$
4: $\tilde{\mathrm{p}}^{(0)} \leftarrow [\frac{1}{|\mathcal{Y}^{\text{test}}|},\dots,\frac{1}{|\mathcal{Y}^{\text{test}}|}]$
5: **for** $t$: $0,\dots,T-1$ **do**
6: $\quad \tilde{\mathrm{p}}_l^{(t+1)} \leftarrow \sum_{l'\in\mathcal{Y}^{\text{test}}} \tilde{\mathrm{p}}_{l'}^{(t)} \cdot a_{ll'}$
7: $\quad \tilde{\mathrm{p}}^{(t+1)} \leftarrow \tilde{\mathrm{p}}^{(t+1)}/\|\tilde{\mathrm{p}}^{(t+1)}\|_1$
8: $\quad$ **if** $\|\tilde{\mathrm{p}}^{(t+1)} - \tilde{\mathrm{p}}^{(t)}\|_1 \leq \varepsilon$ **then**
9: $\quad\quad$ **break**
10: $\quad$ **end if**
11: **end for**
12: **return** $-\log\tilde{\mathrm{p}}^{(t)}$

---

**Algorithm 2** Pipeline of **A**daptive **D**ebiasing **T**sallis **E**ntropy (ADTE) for Test-Time Adaptation (TTA)

1: **Input**: Estimated bias $\mathbf{b}$, test input image $\mathbf{x}^{\text{test}}$, the number of image augmentation $N$, the number of selected confident views $N_v$
2: **Output**: Prediction $\hat{y}$
3: views set $\mathbf{X}^{\text{test}} \leftarrow$ augment($\mathbf{x}^{\text{test}}$, num_views=N)
4: Calculate $\mathbf{P}(\cdot|\mathbf{x}_j^{\text{test}}), \forall \mathbf{x}_j^{\text{test}} \in \mathbf{X}^{\text{test}}$
5: Calculate $q^l$ for each class with Equation 13
6: Calculate $\mathbf{H}_{\text{ADTE}}(\mathbf{P}(\cdot|\mathbf{x}_j^{\text{test}})), \forall \mathbf{x}_j^{\text{test}} \in \mathbf{X}^{\text{test}}$ with Equation 8
7: Confident views set $\hat{\mathbf{X}}^{\text{test}} \leftarrow$ top $N_v$ views with the smallest $\mathbf{H}_{\text{ADTE}}$ values
8: $\tilde{\mathbf{P}} \leftarrow$ aggregate($\{\mathbf{P}(\cdot|\hat{\mathbf{x}}_j^{\text{test}})| \forall \hat{\mathbf{x}}_j^{\text{test}} \in \hat{\mathbf{X}}^{\text{test}}\}$)
9: $\hat{y} \leftarrow \arg\max_l \tilde{\mathbf{P}}(y=l|\cdot)$
10: **return** $\hat{y}$

---

Based on our analysis in Section 4.2, when $0 < q < 1$, TE can alleviate the impact of bias, and a smaller value of $q^l$ results in a greater degree of correction. The magnitude of this correction depends directly on the prediction bias of the category. Therefore, we can deduce the relationship between the prediction bias and the corresponding $q^l$ value for each category:

$$\left.\begin{array}{l} \blacktriangleright \texttt{Head class}\ l^{\text{H}}, \text{dis}[\mathbf{P}_{l^{\text{H}}}, \hat{\mathbf{P}}_{l^{\text{H}}}] > 0, \mathbf{P}_{l^{\text{H}}} \to 1, \mathbf{P}_{l^{\text{H}}} > \hat{\mathbf{P}}_{l^{\text{H}}} \\ \blacktriangleright \texttt{Tail class}\ l^{\text{T}}, \text{dis}[\mathbf{P}_{l^{\text{T}}}, \hat{\mathbf{P}}_{l^{\text{T}}}] > 0, \mathbf{P}_{l^{\text{T}}} \to 0, \mathbf{P}_{l^{\text{T}}} < \hat{\mathbf{P}}_{l^{\text{T}}} \end{array}\right\} \Rightarrow -\mathbf{p}\log(\mathbf{p}) < -\hat{\mathbf{p}}\log(\hat{\mathbf{p}}), \quad (9)$$

where the term $\text{dis}[a,b]$ represents the size of the bias between a biased prediction $a$ and the true unbiased prediction $b$. The size of this bias is inversely proportional to $(-\mathbf{p}\log\mathbf{p}) - (-\hat{\mathbf{p}}\log\hat{\mathbf{p}})$, and as $\text{dis}[\cdot,\cdot]$ increases, the information entropy $(-\mathbf{p}\log\mathbf{p})$ for category $l$ is increasingly underestimated, which requires a smaller $q^l$ for effective correction.

Therefore, the class-specific parameter $q^l$ can be calculated indirectly by estimating the bias for each category. We adopt the bias estimation method from Frolic (Zhu et al., 2024) as follows. First, we estimate the prior probability of each class, $\tilde{\mathrm{p}}_l = \mathbf{P}(l)$, by solving the following linear equation:

$$\tilde{\mathrm{p}}_l = \sum_{l'\in\mathcal{Y}^{\text{test}}} \tilde{\mathrm{p}}_{l'} \cdot \mathbb{E}_{\mathbf{x}\sim\mathbf{P}(\mathbf{x}|l')}[\mathbf{P}(l\mid\mathbf{x})]. \tag{10}$$

A memory bank with size $M$ is maintained to store and update continuously incoming test instances for calculating $\mathbb{E}_{\mathbf{x}\sim\mathbf{P}(\mathbf{x}|l')}[\mathbf{P}(l\mid\mathbf{x})]$ in 10. Since true labels in $\mathcal{Y}^{\text{test}}$ are not available, we use pseudo-labels $\hat{l}(\mathbf{x}) := \arg\max_l \mathbf{P}(y=l\mid\mathbf{x})$ as a substitute, following the approach Frolic (Zhu et al., 2024), *i.e.*:

$$\mathbb{E}_{\mathbf{x}\sim\mathbf{P}(\mathbf{x}|l')}[\mathbf{P}(l\mid\mathbf{x})] = \frac{1}{N_{l'}} \sum_{\mathbf{x}|\hat{l}(\mathbf{x})=l'} \mathbf{P}(l\mid\mathbf{x}), \tag{11}$$

where $\mathbf{P}(l\mid\mathbf{x})$ is predicted by the model and $N_{l'} = \sum_{\mathbf{x}|\hat{l}(\mathbf{x})=l'} 1$. We solve for $\tilde{\mathrm{p}}_l$ using Jacobi iteration, with a uniform initialization $\tilde{\mathrm{p}}_l^{(0)} = \frac{1}{|\mathcal{Y}^{\text{test}}|}, \forall l \in \mathcal{Y}^{\text{test}}$. The iteration is given by:

$$\tilde{\mathrm{p}}_l^{(t+1)} = \sum_{l'\in\mathcal{Y}^{\text{test}}} \tilde{\mathrm{p}}_{l'}^{(t)} \cdot \mathbb{E}_{\mathbf{x}\sim\mathbf{P}(\mathbf{x}|l')}[\mathbf{P}(l\mid\mathbf{x})]. \tag{12}$$

In each iteration, we perform L1 normalization over $\tilde{\mathrm{p}}^{(t)} = [\tilde{\mathrm{p}}_1^{(t)},\dots,\tilde{\mathrm{p}}_{|\mathcal{Y}^{\text{test}}|}^{(t)}]^\top$ so that its L1 norm equals 1. The iterative process terminates when the maximum number of iterations is reached or when the convergence condition $\|\tilde{\mathrm{p}}^{(t+1)} - \tilde{\mathrm{p}}^{(t)}\|_1 \leq \varepsilon$ is met, where $\varepsilon$ is the convergence threshold.

Combining this with Section 4.2, we normalize the estimated bias vector $\tilde{\mathbf{p}}$ for all categories into the interval $(0,1)$. A larger bias $\tilde{\mathrm{p}}_l$ corresponds to a smaller $q^l$. We use min-max normalization to map

the bias values to the parameter range:

$$q^l = \alpha + (\beta - \alpha)\frac{-\log \tilde{p}_l - \min(\tilde{\mathbf{p}})}{\max(\tilde{\mathbf{p}}) - \min(\tilde{\mathbf{p}})}, \tag{13}$$

where $\alpha$ and $\beta$ define the normalization interval. After computing the class-specific parameter $q^l$, we plug it into Equation 8 to calculate $\mathbf{H}_{\texttt{ADTE}}$ for each augmented view, which is then used to select high-confidence views. After obtaining the probability distributions of the selected high-confidence views, we simply average (aggregate) these distributions to produce the final prediction. The algorithm for estimating the bias $\tilde{\mathbf{p}}$ is summarized in Algorithm 1, and the overall TTA pipeline is summarized in Algorithm 2.

## 5 EXPERIMENTS

### 5.1 EXPERIMENTAL SETUP

**Benchmarks.** Two standard benchmarks: (a) Out-of-Distribution (OOD): evaluates generalization to distributions different from training, including ImageNet (Deng et al., 2009) and variants (ImageNet-A (Hendrycks et al., 2019), -V2 (Recht et al., 2019), -R (Hendrycks et al., 2020), and -K (Wang et al., 2019)); (b) Cross-Domain: classification across diverse domains—generic objects (Caltech (Fei-Fei et al., 2005)), scenes (SUN (Xiao et al., 2010)), textures (DTD (Cimpoi et al., 2014)), satellite images (EuroSAT (Helber et al., 2019)), actions (UCF (Soomro et al., 2012)), and five fine-grained datasets (Pets (Parkhi et al., 2012), Cars (Krause et al., 2013), Flowers (Nilsback & Zisserman, 2008), Food (Bossard et al., 2014), Aircraft (Maji et al., 2013)).

**Implementation Details.** Following (Farina et al., 2024; Zhu et al., 2024), we use pretrained CLIP (`ViT-B/16`, `ViT-L/14`), with `ViT-B/16` serving as the default model for ablation studies. For a fair comparison, we use two types of text prompts: *hand-crafted templates* from methods (Shu et al., 2022; Farina et al., 2024; Karmanov et al., 2024; Zhou et al., 2025), and *text descriptions generated by GPTs*, as in CuPL (Pratt et al., 2023; Zhu et al., 2024). We do not tune any hyperparameters, instead adopting the setup from Zero (Farina et al., 2024), with $N = 64$ augmented views and a confidence-based filtering ratio of $0.1$. The memory bank size for each category is set to $10$. The normalization interval for the class-specific parameter $q^l$ is $[0.01, 0.9]$. All experiments were conducted on a single `NVIDIA A100` GPU, with results averaged over 3 seeds.

### 5.2 COMPARISONS WITH STATE-OF-THE-ART

Table 1: Accuracy comparison (%) on ImageNet and its variants for CLIP `ViT-B/16` and `ViT-L/14`.

| | Method | IN | IN-V2 | IN-K | IN-A | IN-R | Average | OOD Avg |
|---|---|---|---|---|---|---|---|---|
| **ViT-B/16** | CLIP [ICML 2022] | 68.7 | 62.2 | 48.3 | 50.6 | 77.7 | 61.5 | 59.7 |
| | TPT [NeurIPS 2022] | 68.9 | 63.4 | 47.9 | 54.7 | 77.0 | 62.4 | 60.8 |
| | TDA [CVPR 2024] | 69.5 | 64.6 | 50.5 | 60.1 | 80.2 | 65.0 | 63.9 |
| | Zero [NeurIPS 2024] | 70.9 | 65.1 | 50.3 | 64.0 | 80.8 | 66.2 | 65.0 |
| | Dyna [ICLR 2025] | 69.6 | 64.7 | 48.2 | 56.2 | 78.2 | 63.4 | 61.8 |
| | BCA [CVPR 2025] | 70.2 | 64.9 | 50.9 | 61.1 | 80.7 | 65.6 | 64.4 |
| | **ADTE**$_{\texttt{Templates}}$ | **71.8** | **65.6** | **53.5** | **65.5** | **81.4** | **67.5** | **66.5** |
| | CuPL [ICCV 2023] | 69.9 | 64.4 | 49.4 | 59.7 | 79.5 | 64.6 | 63.3 |
| | Frolic [NeurIPS 2024] | 70.9 | 64.7 | 53.3 | 60.4 | 80.7 | 66.0 | 64.8 |
| | **ADTE**$_{\texttt{CuPL}}$ | **72.7** | **66.2** | **54.3** | **63.5** | **80.9** | **67.5** | **66.2** |
| **ViT-L/14** | CLIP [ICML 2022] | 75.9 | 70.2 | 59.7 | 70.9 | 87.9 | 72.9 | 72.2 |
| | TPT [NeurIPS 2022] | 75.5 | 70.0 | 59.8 | 74.7 | 87.9 | 73.6 | 73.1 |
| | TDA [CVPR 2024] | 76.3 | 71.5 | 61.3 | 77.9 | 89.8 | 75.4 | 75.1 |
| | Zero [NeurIPS 2024] | 77.2 | 71.9 | 61.1 | 80.7 | 90.2 | 76.2 | 75.9 |
| | **ADTE**$_{\texttt{Templates}}$ | **77.8** | **72.8** | **63.5** | **81.1** | **90.6** | **77.2** | **77.0** |
| | CuPL [ICCV 2023] | 76.2 | 71.9 | 60.7 | 77.9 | 89.6 | 75.3 | 75.0 |
| | Frolic [NeurIPS 2024] | 77.4 | 72.5 | 63.1 | 78.9 | 90.3 | 76.4 | 76.2 |
| | **ADTE**$_{\texttt{CuPL}}$ | **78.2** | **73.3** | **63.9** | **81.0** | **90.4** | **77.4** | **77.2** |

We compare `ADTE` with several state-of-the-art methods on both OOD and cross-domain benchmarks, including CLIP (Radford et al., 2021), TPT (Shu et al., 2022), TDA (Karmanov et al., 2024),

Zero (Farina et al., 2024), Dyna (Xiao et al., 2025), BCA (Zhou et al., 2025), CuPL (Pratt et al., 2023) and Frolic (Zhu et al., 2024). We present results for `ADTE` using both types of text prompts.

**Results on the OOD Benchmark**. In Table 1, `ADTE` consistently outperforms other methods on both `ViT-B/16` and `ViT-L/14` backbones, and with both template-based and text-description-based prompts. On ImageNet-1k, `ADTE`~Templates~ achieves 71.8% accuracy, 0.9% higher than Zero. `ADTE`~CuPL~ reaches 72.7%, outperforming Frolic by 1.8%. For OOD datasets, `ADTE`~Templates~ surpasses state-of-the-art methods on IN-V2 (65.6% vs Zero's 65.1%), IN-K (53.5% vs BCA's 50.9%), IN-A (65.5% vs Zero's 64.0%), and IN-R (81.4% vs Zero's 80.8%). `ADTE`~CuPL~ also consistently outperforms Frolic on all datasets, with a significant 3.1% lead on IN-A. For the `ViT-L/14` backbone, `ADTE`~Templates~ achieves 77.2% average accuracy and 77.0% OOD average accuracy, both 1.1% higher than Zero. Compared to Frolic, `ADTE`~CuPL~ shows the best performance with 77.4% overall average and 77.2% OOD average, confirming its consistent superiority.

Table 2: Accuracy comparison (%) on 10 cross-domain datasets for CLIP `ViT-B/16` and `ViT-L/14`.

| | Method | Pets. | Flow. | Airc. | DTD. | Euro. | Cars. | Food. | SUN. | Calt. | UCF. | Avg. |
|---|---|---|---|---|---|---|---|---|---|---|---|---|
| ViT-B/16 | CLIP [ICML 2022] | 88.9 | 70.4 | 24.8 | 44.3 | 47.7 | 65.2 | 86.1 | 62.5 | 92.9 | 66.7 | 64.9 |
| | TPT [NeurIPS 2022] | 87.7 | 68.9 | 24.7 | 47.7 | 42.4 | 66.8 | 84.6 | 65.5 | 94.1 | 68.0 | 65.0 |
| | TDA [CVPR 2024] | 88.6 | 71.4 | 23.9 | 47.4 | 58.0 | 67.2 | 86.1 | 67.6 | 94.2 | 70.6 | 67.5 |
| | Zero [NeurIPS 2024] | 87.2 | 66.8 | 24.4 | 45.9 | 43.8 | 68.5 | 84.6 | 66.9 | 94.1 | 68.6 | 65.1 |
| | Dyna [ICLR 2025] | 88.3 | 69.9 | 24.3 | 48.0 | 42.3 | 67.7 | 85.4 | 66.3 | 94.3 | 68.7 | 65.5 |
| | BCA [CVPR 2025] | **90.4** | **73.1** | 28.6 | **53.5** | 56.6 | 66.8 | 86.0 | 68.4 | 94.7 | 67.6 | 68.6 |
| | **ADTE**~Templates~ | 89.7 | 72.6 | **28.9** | 49.5 | 53.8 | **70.9** | 86.3 | 70.4 | 94.8 | 73.1 | **69.0** |
| | CuPL [ICCV 2023] | 92.0 | 73.2 | 27.7 | 54.3 | 52.7 | 66.4 | 86.2 | 68.5 | 94.6 | 70.7 | 68.6 |
| | Frolic [NeurIPS 2024] | **92.9** | 74.8 | 31.5 | 56.1 | 58.5 | 69.1 | **87.2** | 70.8 | 95.2 | 75.2 | 71.1 |
| | **ADTE**~CuPL~ | 92.7 | **75.4** | **33.5** | **57.2** | 58.2 | **71.2** | 86.9 | **71.3** | **95.7** | **75.5** | **71.8** |
| ViT-L/14 | CLIP [ICML 2022] | 93.5 | 79.3 | 32.4 | 53.0 | 58.0 | 76.8 | 91.0 | 67.5 | 94.8 | 74.2 | 72.0 |
| | TPT [NeurIPS 2022] | 93.6 | 76.2 | 31.9 | 55.2 | 51.8 | 77.7 | 88.9 | 70.2 | 95.5 | 74.9 | 71.5 |
| | TDA [CVPR 2024] | 93.5 | **80.5** | 34.7 | 56.7 | 64.1 | 78.3 | 90.9 | 71.5 | 95.9 | 76.6 | 74.2 |
| | Zero [NeurIPS 2024] | 93.4 | 79.2 | 33.9 | 53.7 | 54.1 | 78.5 | 90.2 | 72.1 | 96.0 | 77.1 | 72.8 |
| | **ADTE**~Templates~ | **93.7** | 79.4 | **37.2** | **59.7** | 56.2 | **79.4** | **91.8** | **73.2** | **96.4** | **80.6** | **74.8** |
| | CuPL [ICCV 2023] | 94.3 | 79.8 | 35.5 | 62.7 | 61.2 | 78.0 | 91.3 | 72.4 | 96.7 | 75.9 | 74.7 |
| | Frolic [NeurIPS 2024] | **94.9** | 82.4 | 40.0 | 64.1 | **66.2** | 80.8 | **91.8** | 74.5 | 97.2 | 80.0 | 77.1 |
| | **ADTE**~CuPL~ | 94.6 | **83.8** | **40.8** | **65.6** | 65.9 | **81.8** | 91.7 | **74.8** | **97.4** | **80.4** | **77.7** |

**Results on the Cross Domain Benchmark**. Table 2 demonstrates the significant advantage of `ADTE` in cross-domain image recognition. On the `ViT-B/16` backbone, `ADTE`~Templates~ (69.0%) outperforms all template-based methods, while `ADTE`~CuPL~ (71.8%) surpasses Frolic (71.1%). Similarly, on `ViT-L/14`, `ADTE`~Templates~ (74.8%) and `ADTE`~CuPL~ (77.7%) both significantly outperform their respective counterparts, TDA (74.2%) and Frolic (77.1%). `ADTE`~Templates~ also ranks at or near the top on multiple individual datasets. For instance, on `ViT-B/16`, it improves performance by 2.4%, 2.0%, and 2.5% on the Cars, SUN, and UCF datasets, respectively, compared to the best competing method. On `ViT-L/14`, it achieves state-of-the-art results on Aircraft (37.2%, +2.5% improvement over TDA), DTD (59.7%, +3.0% improvement over TDA), and UCF (80.6%, +3.5% improvement over Zero).

## 5.3 Ablation Studies

We evaluate the contributions of our proposed components, `ADTE` and Logit Adjustment (`LA`), through an ablation study. All experiments in this section are conducted on both `ViT-B/16` and `ViT-L/14` architectures, with results shown in Table 3.

**Effectiveness of ADTE.** Removing `ADTE` leads to a notable performance decline across all metrics compared to the full `ADTE`~Templates~ model. For `ViT-B/16`, accuracy drops by 1.1% on IN-Variants, 0.7% on IN, and 2.1% on the 10-datasets. For `ViT-L/14`, accuracy decreases by 0.7%, 0.5%, and 1.6% on the same benchmarks. The performance drop is most significant on the

Table 3: Accuracy (%) of different models on 10-datasets, including ImageNet and its five variant datasets.

| Module | | ViT-B/16 | | | ViT-L/14 | | |
|---|---|---|---|---|---|---|---|
| | | IN-Variants | IN | 10-datasets | IN-Variants | IN | 10-datasets |
| ▲ | ADTE$_{\texttt{Templates}}$ | **66.5** | **71.8** | **69.0** | **77.0** | **77.8** | **74.8** |
| △ | *w/o* **ADTE** | 65.4 | 71.1 | 66.9 | 76.3 | 77.3 | 73.2 |
| △ | *w/o* **Logit Adjustment (LA)** | 66.1 | 71.6 | 68.8 | 76.7 | 77.6 | 74.1 |
| △ | *w/o* **ADTE** and **LA** | 65.0 | 70.9 | 65.1 | 75.9 | 77.2 | 72.8 |

10-datasets ($-2.1\%$ and $-1.6\%$), indicating that ADTE is particularly effective at enhancing the model's generalization across diverse cross-domain tasks.

**Contribution of Logit Adjustment.** LA adjusts the model logits by using bias estimation (Zhu et al., 2024). The experiments show that removing LA also decreases performance, but this decline is generally smaller than removing ADTE. For ViT-B/16, the accuracy drops are $0.4\%$, $0.2\%$, and $0.2\%$, while ViT-L/14 experiences decreases of $0.3\%$, $0.2\%$, and $0.7\%$. Both ADTE and LA contribute positively to the model's accuracy. Furthermore, LA can be seamlessly integrated with ADTE to boost performance without adding extra computational cost.

## 5.4 FURTHER ANALYSIS

**Comparison Among SE, TE, and ADTE**. Table 4 compares the performance of ADTE, SE, and TE, all with logit adjustment (LA). ADTE demonstrates the highest performance. This highlights ADTE's distinct advantages in tasks that rely on entropy information.

Table 4: Results for SE, TE, and ADTE.

| Module | IN | IN-V2 | IN-K | IN-A | IN-R |
|---|---|---|---|---|---|
| LA-SE | 70.7 | 64.1 | 52.3 | 64.0 | 80.2 |
| SE-LA | 71.0 | 64.3 | 52.5 | 64.1 | 80.3 |
| TE-LA (q=0.5) | 71.6 | 65.2 | 53.0 | 64.8 | 80.7 |
| **ADTE** | **71.8** | **65.6** | **53.5** | **65.5** | **81.4** |

Table 5: Computational cost and effect of different intervals.

| Interval | IN | IN-V2 | IN-K | IN-A | IN-R | Avg |
|---|---|---|---|---|---|---|
| $[0.1, 0.9]$ | 71.7 | 65.5 | 53.4 | 65.2 | 81.3 | 67.4 |
| $[\mathbf{0.01}, \mathbf{0.9}]$ | **71.8** | **65.6** | 53.5 | 65.5 | **81.4** | **67.5** |
| $[0.001, 0.9]$ | **71.8** | **65.6** | **53.6** | **65.6** | 81.2 | **67.5** |
| $[0.0001, 0.9]$ | 71.7 | 65.5 | **53.6** | **65.6** | 81.2 | **67.5** |

| Metric | TPT | Zero | ADTE |
|---|---|---|---|
| Time[$s$] | 0.57±0.01 | 0.06±0.01 | 0.07±0.01 |
| Mem[GB] | 17.66 | 1.40 | 1.46 |

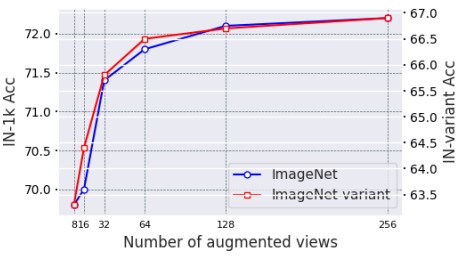

Figure 4: Different number of views.

**Normalization Intervals and Computational Cost.** Table 5 shows the performance ADTE using different normalization intervals. As the lower bound of the interval decreases from 0.1 to 0.0001, performance generally improves and then stabilizes. This demonstrates that ADTE is effective when the class-specific parameter is less than 1 and that the method is robust as it does not require precise parameter tuning; as long as the lower bound is sufficiently small, excellent performance is achieved. Furthermore, ADTE adds only a negligible computational cost compared to Zero, primarily for the bias estimation step.

**Number of Augmented Views**. Figure 4 shows that increasing views ($N$) improves performance until convergence. And more views help ADTE select more confident views, boosting accuracy.

**Memory Bank Size.** We systematically evaluated the impact of varying memory bank sizes on ADTE's performance on ImageNet-1k, with quantitative results presented in Table 6.

Table 6: Impact of memory bank size on performance (%).

| Size | 1 | 2 | 5 | 10 | 20 | 30 | 40 |
|---|---|---|---|---|---|---|---|
| **ADTE** | 71.71 | 71.73 | 71.74 | 71.83 | 71.81 | 71.85 | 71.86 |

The results show that the memory bank size has a minimal impact on ADTE's performance: with a size of 1, the model already achieves $71.71\%$ accuracy. As the size increases to 10, the accuracy

stabilizes around 71.83%. Further expansion to 40 yields only marginal fluctuations (71.86%). A small number of high-confidence samples is sufficient to support reliable adaptive adjustments.

Note that 71.71% accuracy achieved with size 1 does not mean the memory bank can be removed. To see this, consider 200 categories, using a memory bank size of 1 yields 200 samples in total. The statistical information used to calculate bias for a specific category is derived not only from its own sample but from all 200 pseudo-labeled samples in the bank.

**Improvements on `tail` Categories.** To further validate ADTE's effectiveness on `tail` classes, we conducted quantitative experiments on `tail` categories in ImageNet-1k. We selected the last 10 `tail` classes in ImageNet-1k and compared the performance between CLIP and ADTE.

Table 7: Performance improvement on representative tail classes.

| Class Index | 670 | 193 | 981 | 157 | 533 | 168 | 316 | 106 | 50 | 156 |
|---|---|---|---|---|---|---|---|---|---|---|
| Accuracy (CLIP) | 17.4 | 16.7 | 16.1 | 15.0 | 14.5 | 12.5 | 11.5 | 0.00 | 0.00 | 0.00 |
| Confidence (CLIP) | 0.01 | 0.11 | 0.05 | 0.18 | 0.09 | 0.13 | 0.15 | 0.09 | 0.13 | 0.19 |
| **Accuracy (ADTE)** | **16.7** | **33.3** | **33.3** | **38.5** | **25.0** | **59.4** | **35.0** | **32.2** | **41.7** | **56.5** |
| **Confidence (ADTE)** | **0.16** | **0.31** | **0.28** | **0.35** | **0.24** | **0.44** | **0.33** | **0.25** | **0.34** | **0.47** |

As shown in Table 7, ADTE achieves significant improvements in both accuracy and confidence for these categories. For instance, classes that originally had zero accuracy show enhanced accuracy ranging from 32.2% to 56.5% under ADTE, while confidence scores simultaneously increase. This directly validates ADTE's effectiveness in improving `tail` class performance.

Table 8: Comparative analysis between head and tail classes.

| Module | Head (avg_pred) | Head (avg_entropy) | Tail (avg_pred) | Tail (avg_entropy) |
|---|---|---|---|---|
| CLIP | 0.6687 | 5.2956 | 0.1466 | 5.2972 |
| **ADTE** | **0.8112** | **3.2156** | **0.3638** | **3.3761** |

Table 8 presents results of prediction confidence and entropy values for head classes (top 50) and tail classes (bottom 50). The results demonstrate that ADTE not only enhances the average prediction confidence for `tail` classes (from 0.1466 to 0.3638) but also reduces prediction uncertainty by decreasing the entropy value (from 5.2972 to 3.3761). These improvements indicate that the model achieves more definitive and reliable predictions for `tail` categories.

**Effect of Different Normalization Functions**. As shown in Table 9, ADTE's performance is not highly dependent on a specific normalization function. While some methods may offer minor advantages on certain datasets, no single method is universally optimal. This demonstrates the robustness of our approach.

Table 9: Results with different normalizations.

| Normalization | IN | IN-V2 | IN-K | IN-A | IN-R |
|---|---|---|---|---|---|
| Z-Score | 71.8 | 65.9 | **53.5** | 65.5 | 81.3 |
| Decimal Scaling | **72.0** | 65.8 | **53.5** | **65.7** | 81.3 |
| Sigmoid | **72.0** | **66.1** | **53.5** | 65.4 | 81.2 |
| **Min-Max** | 71.8 | 65.6 | **53.5** | 65.5 | **81.4** |

## 6 CONCLUSION AND LIMITATION

**Conclusion.** This paper introduces a novel Test-Time Adaptation (TTA) approach called **A**daptive **D**ebiasing **T**sallis **E**ntropy (ADTE), which is designed to handle the inherent prediction biases in Vision-Language Models (VLMs). We show that Shannon Entropy can be considered a special case of Tsallis Entropy, and that its performance serves as a lower bound. By generalizing Tsallis Entropy with class-specific parameters $q^l$ tailored for each category $l$, ADTE effectively reduces the biases encountered during TTA. Our experimental results demonstrate that ADTE outperforms state-of-the-art methods across multiple datasets, proving its robustness and effectiveness in improving the adaptation performance of VLMs.

**Limitation.** Due to the specific nature of its design, ADTE is highly effective in scenarios with significant prediction bias. However, its advantages may be less apparent in scenarios where the model has almost no biased predictions.

## ACKNOWLEDGEMENTS

This work is supported by the NSFC (62276131, 62506168), Natural Science Foundation of Jiangsu Province of China under Grant (BK20240081, BK20251431).

The authors gratefully acknowledge financial support from the China Scholarship Council (CSC) (Grant No. 202506840036).

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

# Appendix for Adaptive Debiasing Tsallis Entropy for Test-Time Adaptation

## A ADDITIONAL EXPERIMENTS

**Detailed Analysis of Estimated Bias Statistics.** We conduct a statistical analysis of prediction bias on ImageNet variants and cross-domain datasets to explore the correlation between performance gains and the degree of bias. Table 10 shows results on five ImageNet variants.

Table 10: Bias statistical analysis of ImageNet variant datasets.

| Dataset (Classes) | IN-A (200) | IN-R (200) | IN-K (1000) | IN-V (1000) | IN-I (1000) |
|---|---|---|---|---|---|
| Variance | 0.9527 | 0.7650 | 1.5858 | 1.1603 | 1.0833 |
| Std_Dev | 0.9621 | 0.8747 | 1.2953 | 1.0772 | 1.0408 |
| Acc (CLIP) | 50.6 | 77.7 | 48.3 | 62.2 | 68.7 |
| **Acc (ADTE)** | **65.5** | **81.4** | **53.5** | **65.6** | **71.8** |
| Gain | 14.9 | 4.7 | 5.2 | 3.4 | 3.1 |

As shown in Table 10, datasets with larger bias variance and standard deviation (e.g., IN-A and IN-K) correspond to a greater disparity in bias between `head` and `tail` classes. Consequently, the performance gains of `ADTE` are higher on these datasets (14.9% and 5.2%). This indicates that `ADTE` is particularly effective in scenarios with prominent biases, which also explains the variation in performance improvements across different datasets.

For the cross-domain benchmark, we analyzed the average prediction accuracy and confidence across different percentile intervals. As shown in Tables 11 and 12, all cross-domain datasets exhibit significant long-tail distribution characteristics and CLIP demonstrates pronounced prediction bias across them. These results validate the universality of CLIP's prediction bias.

Table 11: Quantitative analysis of prediction bias on cross-domain datasets (Part 1). The values in each cell represent (Accuracy, Confidence).

| Class Percentile | Flower102 | DTD | Pets | Cars | UCF101 |
|---|---|---|---|---|---|
| 0%-10% | (1.0, 0.85) | (0.98, 0.82) | (1.0, 0.95) | (0.99, 0.88) | (1.0, 0.95) |
| 10%-20% | (1.0, 0.91) | (0.86, 0.67) | (0.99, 0.96) | (0.94, 0.81) | (0.98, 0.86) |
| 20%-30% | (0.98, 0.86) | (0.74, 0.59) | (0.98, 0.93) | (0.89, 0.71) | (0.94, 0.80) |
| 30%-40% | (0.93, 0.82) | (0.54, 0.32) | (0.96, 0.87) | (0.82, 0.65) | (0.90, 0.74) |
| 40%-50% | (0.87, 0.69) | (0.42, 0.29) | (0.95, 0.87) | (0.73, 0.51) | (0.82, 0.63) |
| 50%-60% | (0.81, 0.63) | (0.34, 0.23) | (0.93, 0.80) | (0.64, 0.44) | (0.72, 0.55) |
| 60%-70% | (0.62, 0.45) | (0.22, 0.16) | (0.90, 0.81) | (0.56, 0.38) | (0.57, 0.41) |
| 70%-80% | (0.50, 0.35) | (0.07, 0.08) | (0.81, 0.67) | (0.42, 0.34) | (0.32, 0.27) |
| 80%-90% | (0.12, 0.14) | (0.01, 0.04) | (0.63, 0.52) | (0.21, 0.22) | (0.19, 0.19) |
| 90%-100% | (0.00, 0.01) | (0.00, 0.03) | (0.00, 0.00) | (0.06, 0.11) | (0.06, 0.08) |

Table 12: Quantitative analysis of prediction bias on cross-domain datasets (Part 2). The values in each cell represent (Accuracy, Confidence).

| Class Percentile | Caltech101 | Food101 | SUN397 | Aircraft | Eurosat |
|---|---|---|---|---|---|
| 0%-10% | (1.0, 0.96) | (0.96, 0.93) | (0.96, 0.84) | (0.86, 0.66) | (0.82, 0.51) |
| 10%-20% | (1.0, 0.96) | (0.94, 0.87) | (0.90, 0.75) | (0.53, 0.24) | (0.77, 0.63) |
| 20%-30% | (1.0, 0.97) | (0.93, 0.84) | (0.85, 0.69) | (0.37, 0.14) | (0.76, 0.57) |
| 30%-40% | (1.0, 0.97) | (0.91, 0.85) | (0.79, 0.60) | (0.24, 0.12) | (0.72, 0.43) |
| 40%-50% | (1.0, 0.96) | (0.89, 0.82) | (0.72, 0.52) | (0.17, 0.12) | (0.47, 0.35) |
| 50%-60% | (0.98, 0.90) | (0.87, 0.79) | (0.64, 0.47) | (0.10, 0.12) | (0.29, 0.32) |
| 60%-70% | (0.93, 0.87) | (0.85, 0.76) | (0.56, 0.39) | (0.07, 0.07) | (0.15, 0.24) |
| 70%-80% | (0.88, 0.79) | (0.82, 0.72) | (0.47, 0.32) | (0.04, 0.07) | (0.14, 0.13) |
| 80%-90% | (0.81, 0.72) | (0.76, 0.65) | (0.32, 0.24) | (0.00, 0.04) | (0.00, 0.08) |
| 90%-100% | (0.49, 0.42) | (0.64, 0.54) | (0.12, 0.12) | (0.00, 0.03) | (0.00, 0.04) |

**Correlation between $\mathbf{Tcr}_K$ and Prediction Accuracy.** We examine the correlation between $\mathbf{Tcr}_K$ and prediction accuracy on ImageNet-A to validate the reliability of $\mathbf{Tcr}_K$ as an intermediate evaluation metric.

Table 13: Correlation between $\mathbf{Tcr}_K$ and accuracy on ImageNet-A.

| Value of $q$ | $\mathbf{Tcr}_1$ | $\mathbf{Tcr}_3$ | $\mathbf{Tcr}_5$ | $\mathbf{Tcr}_{10}$ | $\mathbf{Tcr}_{20}$ | Accuracy |
|---|---|---|---|---|---|---|
| 0.01 | 29.3069 | 27.0006 | 25.9538 | 24.6061 | 23.7410 | 63.48 |
| 0.1 | 29.3283 | 26.9548 | 25.9067 | 24.5704 | 23.7164 | 63.45 |
| 0.5 | 29.3293 | 26.8951 | 25.8540 | 24.5383 | 23.6985 | 63.24 |
| 0.9 | 29.3189 | 26.8414 | 25.8114 | 24.5162 | 23.6892 | 62.80 |
| 1.1 | 29.3043 | 26.8000 | 25.7808 | 24.5020 | 23.6841 | 62.58 |
| 1.5 | 29.2820 | 26.7511 | 25.7460 | 24.4864 | 23.6785 | 62.68 |
| 2.0 | 29.2682 | 26.7257 | 25.7291 | 24.4798 | 23.6771 | 60.95 |

As shown in Table 13, when keeping $K$ constant while decreasing $q$, both $\mathbf{Tcr}_K$ and accuracy demonstrate an increasing trend. This suggests that $\mathbf{Tcr}_K$ and accuracy can be considered approximately equivalent metrics, as the former evaluates model performance through prediction confidence, while the latter directly measures performance via top-1 prediction correctness.

**Integration with DEYO Method.** We further investigate the modularity of `ADTE` by integrating it with DEYO (Lee et al., 2024b). By simply replacing the Softmax Entropy used in DEYO with our `ADTE` while keeping all other parameters and pretrained weights unchanged, we achieve significant performance improvements. Table 14 demonstrates that the modified method obtains a $2.26\%$ boost

Table 14: Integration with DEYO method on ColoredMNIST.

| Method | Avg Acc (%) | Worst-Group Acc (%) | Time (s) |
|---|---|---|---|
| DEYO | 78.24 | 67.39 | 0.073 |
| DEYO + ADTE | 79.29 | 69.65 | 0.074 |

in Worst-Group Accuracy with almost no additional computational overhead.

**Estimated Bias Stability.** We tracked the changes in the estimated bias's mean and variance as the size of the memory bank increases, with results presented in Table 15. As the size of the memory

Table 15: Stability of estimated bias over memory bank size on ImageNet-1k.

| Size | 1 | 2 | 5 | 10 | 20 | 30 | 40 |
|---|---|---|---|---|---|---|---|
| Mean | -5.2248 | -4.3044 | -4.2567 | -3.8658 | -3.9067 | -3.8224 | -3.8825 |
| Variance | 2.2897 | 2.0415 | 1.7100 | 1.5247 | 1.4618 | 1.4537 | 1.4752 |

bank increases, the variance of the estimated biases gradually decreases. When the size reaches 10, both the mean and the variance tend to stabilize, indicating that only 10 samples per category are needed to reach a stable bias estimation. This trend is consistent with the results of the performance experiment, where the performance also stabilizes around the size of 10. `ADTE` can quickly converge to a reliable bias estimation during the test without relying on an excessively large memory bank.

**Experiments on Corrupted Datasets (ImageNet-C, CIFAR-10-C, CIFAR-100-C).** To further demonstrate the effectiveness of CGPO, we conducted experiments on ImageNet-C, CIFAR-10-C, and CIFAR-100-C datasets (Hendrycks & Dietterich, 2019). For ImageNet-C, we randomly select 3 representative corruption families (Defocus Blur, Glass Blur, Motion Blur). For CIFAR-10-C and CIFAR-100-C, we evaluated the models on 7 diverse corruption types, covering blur, noise, and geometric distortions.

As shown in Tables 16, 17 and 18, across all datasets and nearly all corruption types, ADTE consistently outperforms both the original CLIP and SE. Importantly, (1) SE often degrades performance compared to the CLIP model, especially under severe blur and noise; (2) ADTE remains stable and delivers consistent improvements, demonstrating stronger robustness even in settings where SE becomes unreliable.

Table 16: Accuracy (%) on ImageNet-C across severity levels (1–5).

| Model | defocus_blur | | | | | glass_blur | | | | | motion_blur | | | | |
|---|---|---|---|---|---|---|---|---|---|---|---|---|---|---|---|
| | 1 | 2 | 3 | 4 | 5 | 1 | 2 | 3 | 4 | 5 | 1 | 2 | 3 | 4 | 5 |
| CLIP-ViT/B-16 | 58.78 | 53.82 | 43.36 | 34.42 | 26.12 | 56.47 | 48.27 | 26.75 | 20.88 | 16.96 | 62.51 | 56.91 | 47.46 | 34.37 | 26.55 |
| SE | 58.62 | 54.06 | 43.15 | 34.35 | 26.39 | 55.74 | 46.98 | 26.34 | 19.87 | 15.56 | 62.69 | 57.18 | 46.41 | 33.87 | 25.31 |
| **ADTE** | **59.43** | **54.89** | **44.05** | **35.06** | **26.95** | **57.56** | **48.52** | **27.58** | **21.65** | **17.84** | **63.83** | **58.23** | **48.22** | **34.54** | **27.06** |

Table 17: Average Accuracy (%) on CIFAR-10-C.

| Model | brightness | elastic_transform | gaussian_blur | impulse_noise | motion_blur | shot_noise | speckle_noise |
|---|---|---|---|---|---|---|---|
| CLIP-ViT/B-16 | 90.94 | 84.33 | **90.54** | 87.94 | 87.33 | 84.73 | 85.01 |
| SE | 89.86 | 83.71 | 88.92 | 86.88 | 86.98 | 84.39 | 84.86 |
| **ADTE** | **91.32** | **84.86** | 89.96 | **88.43** | **87.65** | **85.56** | **85.42** |

Table 18: Average Accuracy (%) on CIFAR-100-C.

| Model | brightness | elastic_transform | gaussian_blur | impulse_noise | motion_blur | shot_noise | speckle_noise |
|---|---|---|---|---|---|---|---|
| CLIP-ViT/B-16 | 68.29 | 57.83 | **68.05** | 62.46 | **62.17** | 57.27 | **57.66** |
| SE | 67.92 | 58.05 | 66.56 | 61.78 | 61.32 | 57.01 | 56.22 |
| **ADTE** | **68.80** | **59.32** | 67.68 | **62.65** | 61.98 | **57.56** | 57.43 |

Consistent gains in 3 datasets and most corruption severities indicate the generalization ability of ADTE, reinforcing our main claim: ADTE provides a more robust correction mechanism than SE under distribution shifts, even when SE partially or completely fails

**Analysis of Pseudo-Label Noise Effects on Class-Wise Bias Estimation.** To directly measure the effect of noisy pseudo-labels, we conducted controlled experiments in which we manually inject pseudo-label noise on the ImageNet- A dataset. Specifically, for each sample, with probability $p \in \{20\%, 40\%, 60\%, 80\%, 100\%\}$, the true label is replaced by a random incorrect label. This allows us to isolate the impact of degraded pseudo-label quality under varying noise intensities.

Table 19: ImageNet-A Accuracy (%).

| Model | Zero-shot | True Label | Pseudo Label | 20% Noise | 40% Noise | 60% Noise | 80% Noise | 100% Noise |
|---|---|---|---|---|---|---|---|---|
| CLIP-ViT-B/16 | 50.6 | – | – | – | – | – | – | – |
| SE | 64.0 | – | – | – | – | – | – | – |
| ADTE | – | **65.9** | 65.5 | **65.9** | 65.8 | 65.6 | 65.4 | 64.6 |

The results are shown in Table 19. We observe 3 important trends: (1) When using the true label to estimate bias, ADTE presents the best performance compared with results under pseudo-label noise; (2) ADTE remains extremely stable under moderate and even severe noise levels: accuracy stays within a very narrow band ($65.9 \rightarrow 65.4$) even as pseudo-label corruption increases to 80%; (3) Even with 100% incorrect pseudo-labels, ADTE still outperforms SE (64.6 vs. 64.0).

This aligns precisely with our theoretical analysis in Sections 4.1 and 4.2: SE is a strict lower bound of TE, and TE is a strict lower bound of ADTE, since SE corresponds to the special case where all category-wise parameters $q^l$ are identical. Therefore, ADTE can never perform worse than SE, even under extreme pseudo-label noise.

Overall, ADTE is robust to pseudo-label errors, noise-insensitive, and maintains reliable performance even under worst-case degradation, making it practical for real-world deployment where pseudo-labels are inevitably imperfect.

**Performance under Low-Bias Scenarios.** To quantify the performance gap between ADTE and SE/TE under low-bias conditions, we conducted experiments on ImageNet-1k under progressively lower bias conditions. Specifically, we sort ImageNet-1k classes by their prediction-bias magnitude and randomly construct 5 subsets:

- $s_1$: highest inter-class bias (200 most biased classes)

- $s_2$: ...

- $s_5$: lowest bias (200 least biased classes)

This setup directly evaluates ADTE, TE, and SE in increasingly unbiased distributions. The results in Table 20 show that: (1) ADTE consistently outperforms SE and TE at all bias levels; (2) The advantage grows as the bias becomes larger; (3) Even in nearly unbiased settings, ADTE still yields notable gains. This quantitatively validates ADTE's applicability limit and supports the theoretical claim of lower-bound.

Table 20: ImageNet-A accuracy (%) across progressively lower bias conditions.

| Model | $s_1$ | $s_2$ | $s_3$ | $s_4$ | $s_5$ |
|---|---|---|---|---|---|
| CLIP-ViT/B-16 | 69.5 | 58.8 | 52.3 | 74.2 | 76.3 |
| SE | 70.4 | 60.3 | 53.1 | 75.9 | 77.8 |
| ADTE | **73.8** | **62.3** | **55.2** | **77.1** | **78.5** |

**Continual or Gradual Domain-Shift Scenarios.** To evaluate ADTE under realistic evolving distributions, we conducted new experiments by mixing five ImageNet variants, i.e., ImageNet-1k, ImageNet-A, ImageNet-V, ImageNet-K, and ImageNet-R, in a fully randomized interleaved stream. This simulates an online TTA scenario where the domain changes unpredictably from sample to sample, making it one of the most challenging continual shift settings.

Table 21: Accuracy (%) on randomized, mixed five ImageNet variants.

| Model | ImageNet-1k | ImageNet-A | ImageNet-V | ImageNet-K | ImageNet-R |
|---|---|---|---|---|---|
| ADTE | 71.8 | 65.5 | 65.6 | 53.5 | 81.4 |
| ADTE_random | 72.0 | 65.8 | 65.4 | 53.5 | 81.2 |

The results in Table 21 demonstrate: (1) ADTE consistently outperforms SE and the CLIP baseline in all domains; (2) No signs of instability or degradation, even when the domain identity changes every few samples; (3) Memory-based bias estimation remains effective because only a small class-wise bank is maintained, which is naturally robust to cross-domain mixing.

**Evaluating ADTE's Applicability Unimodal ImageNet Models and CLIP Successors like SigLIP.** To verify the effectiveness of ADTE on models other than CLIP, we first conducted additional experiments on the ImageNet-A dataset on unimodal ImageNet-pretrained models, following the MEMO (Zhang et al., 2022) test-time adaptation benchmark. As shown in Table 22, ADTE consistently improves performance on ResNext-101, outperforming standard TTA and other adaptation techniques. The results demonstrate that ADTE is not tied to multimodal encoders, and it remains effective on purely vision-based architectures.

Table 22: ImageNet-A Error (%).

| Method | Error(%) | Method | Error(%) |
|---|---|---|---|
| ResNext-101 (baseline) | 90.0 | WSL | 54.9 |
| + TTA | 83.2 | + TTA | 49.1 |
| + Single-point BN | 88.8 | + Single-point BN | 58.9 |
| + MEMO | 84.3 | + MEMO | 43.2 |
| **+ ADTE** | **81.5** | **+ ADTE** | **41.1** |

Next, we conduct evaluations on five generalization models of CLIP. As shown in Table 23, across all models, including SigLIP and SigLIP2, ADTE provides consistent improvements, confirming its robustness and wide applicability.

The above experiments confirm: (1) ADTE generalizes across architectures (unimodal or multimodal); (2) ADTE generalizes across training regimes (contrastive $\rightarrow$ sigmoid loss); (3) ADTE adapts robustly, even for models with stronger native calibration, such as SigLIP2.

**Confidence Intervals or Significance Tests..** To verify whether the improvements are statistically or practically meaningful, we add more runs to get the confidence intervals of accuracy. Table 24 demonstrates the statistical relevance of our improvements.

Table 23: ImageNet-1k Accuracy (%) on Generalization Models with ADTE.

| Model | ImageNet-1k Acc. (%) | +ADTE |
|---|---|---|
| CLIP | 68.7 | **71.8** |
| OpenCLIP | 70.2 | **73.5** |
| EVA-CLIP | 74.7 | **77.4** |
| SigLIP | 76.2 | **78.9** |
| SigLIP2 | 78.2 | **80.1** |

Table 24: Accuracy (%) 95% confidence interval of the results in Table 3.

| Model | ImageNet-1k | ImageNet-A | ImageNet-V | ImageNet-K | ImageNet-R |
|---|---|---|---|---|---|
| CLIP-ViT/B-16 | 68.7 | 50.6 | 62.2 | 48.3 | 77.7 |
| SE | 70.9 | 64.0 | 65.1 | 50.3 | 80.8 |
| ADTE | 71.8 | 65.5 | 65.6 | 53.5 | 81.4 |
| ADTE_95% con. int. | $71.78 \pm 0.44$ | $65.72 \pm 0.50$ | $65.67 \pm 0.17$ | $53.59 \pm 0.25$ | $81.28 \pm 0.25$ |

## B PROOF

### B.1 PROOF OF PROPERTY 1

▶ Shannon-Tsallis $q \to 1$ Equivalence:

**Property 1.** *As $q \to 1$, Tsallis Entropy becomes equivalent to Shannon Entropy, i.e.,*

$$\lim_{q \to 1} \mathbf{H}_{\text{TE}}(\mathbf{P}(\cdot \mid \mathbf{x}_j^{\text{test}})) = \mathbf{H}_{\text{SE}}(\mathbf{P}(\cdot \mid \mathbf{x}_j^{\text{test}})). \tag{14}$$

*Proof.* For Tsallis entropy, we have:

$$\mathbf{H}_{\text{TE}}(\mathbf{P}(\cdot \mid \mathbf{x}_j^{\text{test}})) = \frac{1}{1-q} \left( \sum_{l=1}^{L} \mathbf{P}(y = l \mid \mathbf{x}_j^{\text{test}})^q - 1 \right) = \frac{1}{1-q} \left( \sum_{l=1}^{L} \mathbf{P}(y = l \mid \mathbf{x}_j^{\text{test}})^q - \sum_{l=1}^{L} \mathbf{P}(y = l \mid \mathbf{x}_j^{\text{test}}) \right)$$
$$= \sum_{l=1}^{L} \mathbf{P}(y = l \mid \mathbf{x}_j^{\text{test}}) \left( \frac{\mathbf{P}(y = l \mid \mathbf{x}_j^{\text{test}})^{q-1} - 1}{1-q} \right). \tag{15}$$

The limitation $\lim_{q \to 1}$ only applies to the terms containing $q$. Thus:

$$\lim_{q \to 1} \mathbf{H}_{\text{TE}}(\mathbf{P}(\cdot \mid \mathbf{x}_j^{\text{test}})) = \sum_{l=1}^{L} \mathbf{P}(y = l \mid \mathbf{x}_j^{\text{test}}) \left( \lim_{q \to 1} \frac{\mathbf{P}(y = l \mid \mathbf{x}_j^{\text{test}})^{q-1} - 1}{1-q} \right). \tag{16}$$

We use L'Hopital's rule to solve the limit in the expression:

$$\lim_{q \to 1} \frac{\mathbf{P}(y = l \mid \mathbf{x}_j^{\text{test}})^{q-1} - 1}{1-q} = \lim_{\gamma \to 0} \frac{\mathbf{P}(y = l \mid \mathbf{x}_j^{\text{test}})^{\gamma} - 1}{-\gamma} = \lim_{\gamma \to 0} \frac{\mathbf{P}(y = l \mid \mathbf{x}_j^{\text{test}})^{\gamma} \log \mathbf{P}(y = l \mid \mathbf{x}_j^{\text{test}})}{-1}$$
$$= \frac{1 \times \log \mathbf{P}(y = l \mid \mathbf{x}_j^{\text{test}})}{-1} = -\log \mathbf{P}(y = l \mid \mathbf{x}_j^{\text{test}}), \tag{17}$$

where the second equality uses L'Hopital's rule. Substituting the above into Equation 16, we have:

$$\lim_{q \to 1} \mathbf{H}_{\text{TE}}(\mathbf{P}(\cdot \mid \mathbf{x}_j^{\text{test}})) = \sum_{l=1}^{L} \mathbf{P}(y = l \mid \mathbf{x}_j^{\text{test}}) \left( -\log \mathbf{P}(y = l \mid \mathbf{x}_j^{\text{test}}) \right)$$
$$= -\sum_{l=1}^{L} \mathbf{P}(y = l \mid \mathbf{x}_j^{\text{test}}) \log \mathbf{P}(y = l \mid \mathbf{x}_j^{\text{test}}) \tag{18}$$
$$= \mathbf{H}_{\text{SE}}(\mathbf{P}(\cdot \mid \mathbf{x}_j^{\text{test}})).$$

Therefore, we prove Property 1.

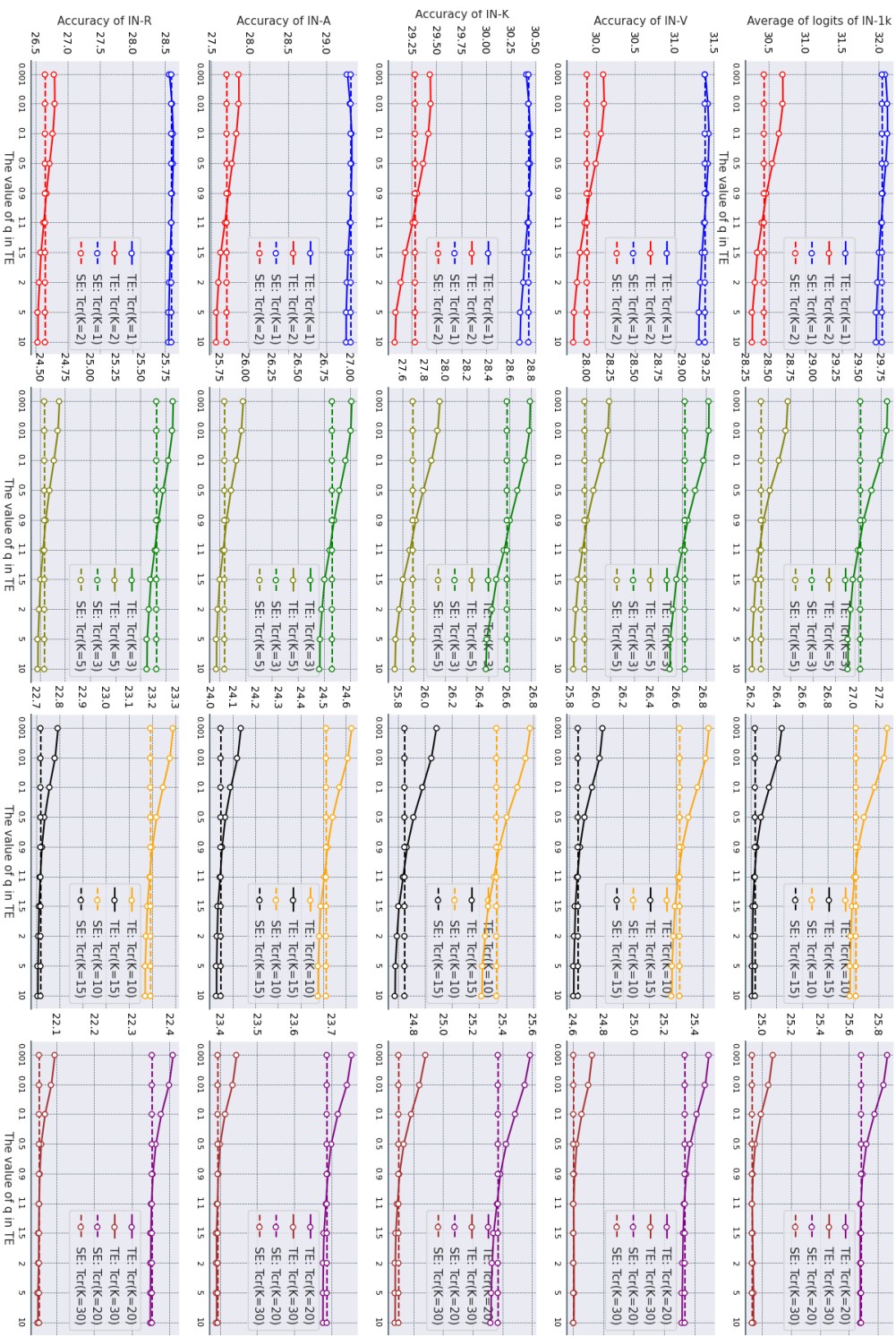

Figure 5: Average $\mathbf{Tcr}_K$ values for different $q$ of TE and SE on ImageNet-1K and its variant datasets.

## B.2 EXPERIMENTAL ANALYSIS OF PROPERTY 2

▶ Higher $\mathbf{Tcr}_K$ under TE as $q \searrow$ and Comparison with SE:

**Property 2.** *Through experimental analysis, we find that as the parameter $q$ decreases, the set of high-confidence views selected by TE tends to have a higher average $\mathbf{Tcr}_K$ value (for $K > 1$). For two different parameter values $q_1$ and $q_2$ with $q_1 < q_2$, and their corresponding selected view sets $\mathcal{X}_{q_1}$ and $\mathcal{X}_{q_2}$ of equal size, we observe:*

$$\frac{1}{|\mathcal{X}_{q_1}|} \sum_{\mathbf{x_1} \in \mathcal{X}_{q_1}} \mathbf{Tcr}_K(\mathbf{x_1}) > \frac{1}{|\mathcal{X}_{q_2}|} \sum_{\mathbf{x_2} \in \mathcal{X}_{q_2}} \mathbf{Tcr}_K(\mathbf{x_2}). \tag{19}$$

*Furthermore, there exists a particular $q^*$ and corresponding view set $\mathcal{X}_{q^*}^{\mathrm{TE}}$, TE outperforms SE in this regard, with the view set selected by SE denoted as $\mathcal{X}^{\mathrm{SE}}$:*

$$\frac{1}{|\mathcal{X}_{q^*}^{\mathrm{TE}}|} \sum_{\mathbf{x_1} \in \mathcal{X}_{q^*}^{\mathrm{TE}}} \mathbf{Tcr}_K(\mathbf{x_1}) > \frac{1}{|\mathcal{X}^{\mathrm{SE}}|} \sum_{\mathbf{x_2} \in \mathcal{X}^{\mathrm{SE}}} \mathbf{Tcr}_K(\mathbf{x_2}). \tag{20}$$

*Experimental analysis.* Equation 19 describes that when $K > 1$, as the parameter $q$ of TE decreases, the selected confidence views exhibit higher average $\mathbf{Tcr}_K$ values. For example, in Figure 5 of ImageNet-1k and its five variants, as the $q$ value decreases from 10 to 0.001, except for $\mathbf{Tcr}(K = 1)$, all other $\mathbf{Tcr}$ values show a gradually increasing trend. This implies that the lower the $q$ value, the more the $TE$ tends to select views with higher similarity scores.

As shown in Figure 5, when the $q$ value of TE is relatively large, the $\mathbf{Tcr}$ value of TE is generally smaller than that of SE. As the $q$ value decreases, the $\mathbf{Tcr}$ value of TE gradually exceeds that of SE. This further indicates that SE is a special case of TE, and appropriate selection of $q$ values can make TE exhibit better performance than SE.

## B.3 PROOF OF CONCLUSIONS 4.2

Let $\mathbf{p} \in (0, \epsilon)$, where $\epsilon$ ensures the probability of $\mathbf{p}$ of tail category being close to zero, and we have the following conclusions:

▶ (1) As $q \to \infty$, we have $\lim_{\mathbf{p} \to 0^+} \mathbf{F}(\mathbf{p}, q) = 0^-$, and $\mathbf{F}(\mathbf{p}, q_1) < \mathbf{F}(\mathbf{p}, q_2) < 0, \ \forall q_1 < q_2 < \infty$.

*Proof.* a. Find the limit of the function $\mathbf{F}(\mathbf{p}, q) = \frac{\mathbf{p}^q}{1-q} + \mathbf{p} \log \mathbf{p}$ as $q \to +\infty$ and $p \to 0^+$.

First term: $\frac{\mathbf{p}^q}{1-q} \to 0^-$ as $q \to +\infty$ and $\mathbf{p} \to 0^+$.

Second term: $\mathbf{p} \log \mathbf{p} \to 0^-$ by L'Hospital's rule: $\lim_{\mathbf{p} \to 0^+} \frac{\log \mathbf{p}}{1/\mathbf{p}} = \lim_{\mathbf{p} \to 0^+} (-\mathbf{p}) = 0^-$

Conclusion:

$$\lim_{\substack{q \to +\infty \\ \mathbf{p} \to 0^+}} \mathbf{F}(\mathbf{p}, q) = 0^- + 0^- = 0^- \tag{21}$$

b. Compare the magnitudes of $\mathbf{F}(\mathbf{p}, q_1)$, $\mathbf{F}(\mathbf{p}, q_2)$, and 0 as $q_1 < q_2 \to +\infty$ and $\mathbf{p} \to 0^+$.

Sign analysis: For $\mathbf{p} \in (0, \epsilon)$ and $q > 1$:

$$\mathbf{F}(\mathbf{p}, q) = \underbrace{\frac{\mathbf{p}^q}{1-q}}_{<0} + \underbrace{\mathbf{p} \log \mathbf{p}}_{<0} < 0 \tag{22}$$

Monotonicity: Fix $\mathbf{p} \in (0, \epsilon)$. The derivative of $\mathbf{G}(q) = \frac{\mathbf{p}^q}{1-q}$:

$$\mathbf{G}'(q) = \frac{\mathbf{p}^q[1 + (1-q)\log \mathbf{p}]}{(1-q)^2} > 0 \quad \text{for } q \to +\infty \tag{23}$$

Thus $\mathbf{G}(q)$ is increasing. For $q_1 < q_2$:

$$\mathbf{G}(q_1) < \mathbf{G}(q_2) \implies \mathbf{F}(\mathbf{p}, q_1) < \mathbf{F}(\mathbf{p}, q_2) < 0 \tag{24}$$

▶ (2) As $q \to 1^+$, we have $\lim_{\mathbf{p} \to 0^+} \mathbf{F}(\mathbf{p}, q) = -\infty$, and $\mathbf{F}(\mathbf{p}, q_1) < \mathbf{F}(\mathbf{p}, q_2) < 0$, $\forall 1 < q_1 < q_2$.

*Proof.* a. Find the limit of the function $\mathbf{F}(\mathbf{p}, q) = \frac{\mathbf{p}^q}{1-q} + \mathbf{p} \log \mathbf{p}$ as $q \to 1^+$ and $p \to 0^+$.

Consider the limit of the entire function $\mathbf{F}(\mathbf{p}, q)$: Since there are two variables approaching their limits, the path of the limit may affect the result. We consider the iterated limits:

① First, fix $q > 1$ and let $\mathbf{p} \to 0^+$. At this time, we have the $\lim_{\mathbf{p} \to 0^+} \mathbf{F}(\mathbf{p}, q) = \lim_{\mathbf{p} \to 0^+} \left( \frac{\mathbf{p}^q}{1-q} + \mathbf{p} \log \mathbf{p} \right)$. Since $q > 1$ is fixed, $\lim_{\mathbf{p} \to 0^+} \frac{\mathbf{p}^q}{1-q} = \frac{0^q}{1-q} = 0^-$, and we have already found that $\lim_{\mathbf{p} \to 0^+} \mathbf{p} \log \mathbf{p} = 0^-$. So, $\lim_{\mathbf{p} \to 0^+} \mathbf{F}(\mathbf{p}, q) = 0^- + 0^- = 0^-$. Then, let $q \to 1^+$, then $\lim_{q \to 1^+}(0^-) = 0^-$.

② Next, fix $\mathbf{p} \in (0, \epsilon)$ and let $q \to 1^+$. For $\lim_{q \to 1^+} \mathbf{F}(\mathbf{p}, q) = \lim_{q \to 1^+} \left( \frac{\mathbf{p}^q}{1-q} + \mathbf{p} \log \mathbf{p} \right)$. When $q \to 1^+$, $1 - q \to 0^-$, $\mathbf{p}^q \to \mathbf{p}$, so $\lim_{q \to 1^+} \frac{\mathbf{p}^q}{1-q} = \frac{\mathbf{p}}{0^-} = -\infty$ (because $\mathbf{p} > 0$), and for a fixed $\mathbf{p} \in (0, \epsilon)$, $\mathbf{p} \log \mathbf{p}$ is a fixed negative value. So, $\lim_{q \to 1^+} \mathbf{F}(\mathbf{p}, q) = -\infty + \mathbf{p} \log \mathbf{p} = -\infty$. Then, let $\mathbf{p} \to 0^+$, then $\lim_{\mathbf{p} \to 0^+}(-\infty) = -\infty$.

Since the results of the two iterative limits are different (one is $0$ and the other is $-\infty$), strictly speaking, the existence of the simultaneous limit depends on the relative rates at which $\mathbf{p}$ and $q - 1$ approach $0$. However, in this work, the probability of the `tail` category is usually a very small decimal, but it will not approach $0^+$ infinitely, and we give priority to the case of $q \to 1^+$. Therefore, $\lim_{q \to 1^+} \mathbf{F}(\mathbf{p}, q) = -\infty + \mathbf{p} \log \mathbf{p} = -\infty$. Therefore, we determine the limit value to be $-\infty$.

b. Compare the magnitudes of $\mathbf{F}(\mathbf{p}, q_1)$, $\mathbf{F}(\mathbf{p}, q_2)$, and $0$ as $q_1 < q_2 \to 1^+$ and $\mathbf{p} \to 0^+$.

This part of the proof is the same as part b of Conclusion (1).

▶ (3) As $q \to 0^+$, we have $\lim_{\mathbf{p} \to 0^+} \mathbf{F}(\mathbf{p}, q) = 1^-$, and $\mathbf{F}(\mathbf{p}, q_1) > \mathbf{F}(\mathbf{p}, q_2) > 0$, $\forall 0 < q_1 < q_2$.

*Proof.* a. Find the limit of the function $\mathbf{F}(\mathbf{p}, q) = \frac{\mathbf{p}^q}{1-q} + \mathbf{p} \log \mathbf{p}$ as $q \to 1^+$ and $p \to 0^+$.

In this condition, both $\mathbf{p}$ and $q$ approach $0$ simultaneously. This is a limit problem for a two-variable function. We need to examine path dependence.

Consider different paths $(\mathbf{p}, q) \to (0^+, 0^+)$:

① First: Iterated limit with $q \to 0^+$ first, then $\mathbf{p} \to 0^+$.

$$\lim_{q \to 0^+} \mathbf{F}(\mathbf{p}, q) = \lim_{q \to 0^+} \left[ \frac{\mathbf{p}^q}{1-q} + \mathbf{p} \log \mathbf{p} \right] = 1^- + \mathbf{p} \log \mathbf{p}. \tag{25}$$

Thus,

$$\lim_{\mathbf{p} \to 0^+} \left( 1^- + \mathbf{p} \log \mathbf{p} \right) = 1^-. \tag{26}$$

② Next: Iterated limit with $\mathbf{p} \to 0^+$ first, then $q \to 0^+$. For fixed $q \in (0, 1)$:

$$\lim_{\mathbf{p} \to 0^+} \mathbf{F}(\mathbf{p}, q) = \lim_{\mathbf{p} \to 0^+} \left[ \frac{\mathbf{p}^q}{1-q} + \mathbf{p} \log \mathbf{p} \right] = 0. \tag{27}$$

Therefore,

$$\lim_{q \to 0^+} (0) = 0. \tag{28}$$

In this work, $\mathbf{p}$ represents the probability of a category, which may be a very small decimal, but it will not approach $0$ infinitely. `Path 2` can be ignored. Thus, we determine the limit value to be $1^-$.

b. Compare the magnitudes of $\mathbf{F}(\mathbf{p}, q_1)$, $\mathbf{F}(\mathbf{p}, q_2)$, and 0 as $q_1 < q_2 \to 0^+$ and $\mathbf{p} \to 0^+$.

`Sign Analysis`: As $\mathbf{p} \to 0^+$, $\mathbf{p} \log \mathbf{p} \to 0^-$. For positivity:

$$\mathbf{F}(\mathbf{p}, q) > 0 \iff \frac{\mathbf{p}^q}{1-q} > -\mathbf{p} \log \mathbf{p} \iff \frac{\mathbf{p}^{q-1}}{1-q} > |\log \mathbf{p}|. \tag{29}$$

As $q \to 0^+$, $q - 1 \approx -1$ and $1 - q \approx 1$. The inequality simplifies to: $\frac{1}{\mathbf{p}} > |\log \mathbf{p}|$. Since $\frac{1}{\mathbf{p}}$ diverges faster than $|\log \mathbf{p}|$ as $\mathbf{p} \to 0^+$, $\mathbf{F}(\mathbf{p}, q) > 0$ for sufficiently small $\mathbf{p}$. Thus:

$$\mathbf{F}(\mathbf{p}, q_1) > 0 \quad \text{and} \quad \mathbf{F}(\mathbf{p}, q_2) > 0. \tag{30}$$

`Monotonicity in` $q$: Fix $\mathbf{p}$. The derivative w.r.t. $q$ is:

$$\mathbf{G}'(q) = \frac{\mathbf{p}^q [1 + (1-q) \log \mathbf{p}]}{(1-q)^2}. \tag{31}$$

For $\mathbf{p} \to 0^+$, $\log \mathbf{p} \to -\infty$. Since $1 - q \approx 1$, the numerator is negative, implying $\mathbf{G}'(q) < 0$. Thus, $\mathbf{F}(\mathbf{p}, q)$ is decreasing in $q$. For $q_1 < q_2$:

$$0 < \mathbf{F}(\mathbf{p}, q_2) < \mathbf{F}(\mathbf{p}, q_1). \tag{32}$$

▶ (4) As $q \to 1^-$, we have $\lim_{\mathbf{p} \to 0^+} \mathbf{F}(\mathbf{p}, q) = +\infty$, and $\mathbf{F}(\mathbf{p}, q_2) > \mathbf{F}(\mathbf{p}, q_1) > 0$, $\forall q_1 < q_2 < 1$.

*Proof.* a. Find the limit of the function $\mathbf{F}(\mathbf{p}, q) = \frac{\mathbf{p}^q}{1-q} + \mathbf{p} \log \mathbf{p}$ as $q \to 1^-$ and $p \to 0^+$.

`Compute the iterated limits`:

① For fixed $q \in (0, 1)$:

$$\lim_{\mathbf{p} \to 0^+} \mathbf{F}(\mathbf{p}, q) = \lim_{\mathbf{p} \to 0^+} \left[ \frac{\mathbf{p}^q}{1-q} + \mathbf{p} \log \mathbf{p} \right] = 0^+ + 0^- = 0^+. \tag{33}$$

Thus,

$$\lim_{q \to 1^-} \left( 0^+ \right) = 0^+. \tag{34}$$

② For fixed $\mathbf{p} \in (0, \epsilon)$:

$$\lim_{q \to 1^-} \mathbf{F}(\mathbf{p}, q) = \lim_{q \to 1^-} \left[ \frac{\mathbf{p}^q}{1-q} + \mathbf{p} \log \mathbf{p} \right] = +\infty + \mathbf{p} \log \mathbf{p} = +\infty. \tag{35}$$

In this work, the probability of the `tail` category is usually a very small decimal, but it will not approach $0^+$ infinitely, and we give priority to the case of $\mathbf{q} \to 1^-$. Therefore, $\lim_{q \to 1^-} \mathbf{F}(\mathbf{p}, q) = +\infty$. Therefore, we determine the limit value to be $+\infty$.

b. Compare the magnitudes of $\mathbf{F}(\mathbf{p}, q_1)$, $\mathbf{F}(\mathbf{p}, q_2)$, and 0 as $q_1 < q_2 \to 1^-$ and $\mathbf{p} \to 0^+$.

`Sign analysis`: Identical to part sign analysis of Conclusion (3).

`Comparing` $\mathbf{F}(\mathbf{p}, q_1)$ `and` $\mathbf{F}(\mathbf{p}, q_2)$: Compute the partial derivative:

$$\frac{\partial \mathbf{F}}{\partial q} = \frac{\mathbf{p}^q}{1-q} \left[ \log \mathbf{p} + \frac{1}{1-q} \right]. \tag{36}$$

Since $\frac{\mathbf{p}^q}{1-q} > 0$, the sign of $\frac{\partial \mathbf{F}}{\partial q}$ depends on $\left[ \log \mathbf{p} + \frac{1}{1-q} \right]$. As $q \to 1^-$, $\frac{1}{1-q} \to +\infty$ dominates $\log \mathbf{p} \to -\infty$. For example:

$$\mathbf{p} = 0.01 \implies \log \mathbf{p} \approx -4.6; \quad q = 0.999 \implies \frac{1}{1-q} = 1000 \implies \log \mathbf{p} + \frac{1}{1-q} > 0.$$

Thus, $\mathbf{F}(\mathbf{p}, q)$ increases with $q$ near $q = 1$. For $q_1 < q_2 \to 1^-$, we have:

$$0 < \mathbf{F}(\mathbf{p}, q_1) < \mathbf{F}(\mathbf{p}, q_2). \tag{37}$$

