# OpenReview forum: "Adaptive Debiasing Tsallis Entropy for Test-Time Adaptation"
_ICLR.cc/2026/Conference — ICLR 2026 Poster_

### Official Review · Reviewer_xmGS · 2025-10-27

**Soundness:** 2
**Presentation:** 2
**Contribution:** 2
**Rating:** 4
**Confidence:** 4

**Summary:**

The paper proposes replacing Shannon entropy (SE) with Tsallis entropy (TE) for selecting confident views in test-time adaptation (TTA), arguing TE better handles prediction bias in vision-language models. It further introduces Adaptive Debiasing Tsallis Entropy (ADTE), which assigns a class-specific parameter $q_l$, derived from an estimated class-prior/bias via a memory bank and pseudo-labels, then picks views with the smallest ADTE to aggregate predictions.

**Strengths:**

1. The paper offers a fresh perspective by replacing Shannon entropy with Tsallis entropy and further adapting a class-specific parameter $q_l$, giving a tunable lens on confidence that can better reflect skewed priors and domain shift during test time.

1. The entropy perspective is modality-agnostic and should, in principle, transfer to other test-time settings (e.g., VLMs, audio, or tabular) where confidence shaping under shift is crucial.

1. The experiments span multiple datasets and architectures, where ADTE outperforms most of them.

**Weaknesses:**

1. Table 3 show that “w/o ADTE and w/o LA” already achieves high accuracy; many reported deltas are modest ($\sim$0.5–2%); without confidence intervals or paired significance tests, it is unclear whether improvements are statistically or practically meaningful.

1. The core of ADTE is a min-max mapping from an estimated bias vector $\tilde{p}$ into $q_l \in [\alpha, \beta]$ (the paper uses $[0.01, 0.9]$) via Eq.~(13). There’s no theoretical link that this linear rescaling is optimal (or even monotone-calibrated) for view selection risk; it feels ad-hoc and dataset-dependent. Sensitivity is only probed by sweeping the lower bound, not the mapping itself.

1. Estimating class priors from pseudo-labels assumes pseudo-labels are reliable; errors in early iterations can feed back into the prior and $q_l$, potentially harming tail classes the method aims to help.

1. The paper did not specify the details of augmentation strategies; which must highly affect the TTA performance.

1. The use of large memory bank (1~10 samples per each of 1000 classes) might be unfair for comparison with baseline TTA methods.

**Questions:**

1. What is the design rationale of min-max mapping?

1. How sensitive is ADTE to pseudo-label noise in the early phase of adaptation? One experiment could be manually altering the class accuracy and see how ADTE behaves.

1. If we replace ADTE with a fixed q or with Shannon entropy but keep the same view-selection pipeline, how much of the reported gain would remain?

---

> ### Author Response · Authors · 2025-11-21
> **Official Response by Authors (1/3)**
>
> Thanks for your valuable suggestions, we will try to address your concerns and we are eager to engage in a more detailed discussion with you.
>
> `Weakness 1. Statistical and Practical Significance of Modest Performance Gains in Table 3 without Confidence Intervals or Significance Tests.`
> - The “w/o ADTE and w/o LA” setting in Table 3 actually corresponds to using Zero as the baseline. As shown in Tables 1 and  2, Zero is already a strong TTA baseline, outperforming most existing methods. Therefore, the fact that ADTE can still achieve additional performance gains on top of Zero further validates its effectiveness.
>
> | Model                     | ImageNet-1k | ImageNet-A | ImageNet-V | ImageNet-K | ImageNet-R |
> |-|-|-|-|-|-|
> | CLIP-ViT/B-16                      | 68.7        | 50.6       | 62.2       | 48.3       | 77.7       |
> | SE                      | 70.9        | 64.0       | 65.1       | 50.3       | 80.8       |
> | ADTE                      | 71.8        | 65.5       | 65.6       | 53.5       | 81.4       |
> | ADTE (95% confidence interval) | 71.78 ± 0.44 | 65.72 ± 0.50 | 65.67 ± 0.17 | 53.59 ± 0.25 | 81.28 ± 0.25 |
> - We also agree with the reviewer’s comment regarding the lack of statistical significance analysis. We have supplemented our work with experiments using multiple random seeds, reported confidence intervals to verify that these improvements are statistically and practically meaningful.
>
> `Weakness 2. Concerns over the Theoretical Justification, Calibration, and Dataset-Dependence of ADTE’s Min–Max Bias Mapping to [α, β], with Limited Sensitivity Analysis on the Mapping Itself.`
> - We expect larger bias values to correspond to smaller $q^l$ values. This is because, as proven in Section 4.2, positive debiasing effects occur only when $0 < q < 1$, and the smaller the $q^l$, the stronger the debiasing effect.
> - Min–max normalization achieves this in the following way: (1) it preserves the relative ordering (monotonicity) of biases across categories; and (2) it constrains $q^l$ within a reasonable, theoretically grounded interval, such as $[0.01, 0.9]$, which satisfies $0 < q < 1$.
> - We do not claim that this mapping is optimal. In fact, as shown in Appendix Table 7, we evaluate different normalization functions (Z-score, decimal scaling, sigmoid function, min–max normalization), and the results are very close (difference < $0.2\\%$). This indicates that as long as the bias is normalized to the range $(0, 1)$ and the relative ordering is preserved, ADTE is not sensitive to the specific mapping function.
> - We agree that the mapping design is replaceable. As suggested by the reviewers, we will add a sensitivity analysis for other monotonic mappings (e.g., logarithmic mapping, softmax mapping) in the subsequent version.

---

> ### Author Response · Authors · 2025-11-21
> **Official Response by Authors (2/3)**
>
> `Weakness 3. Potential Harm to Tail Classes from Error Propagation in Class Prior and $q^l$ Estimation When Pseudo-Labels Are Unreliable.`
> - Pseudo-labels are always generated by the original CLIP model, and CLIP’s parameters are never updated during TTA. Therefore, in both the early and late stages of TTA, pseudo-labels are produced by the same fixed CLIP model.
> - In the early stage of TTA, the memory bank may contain few samples, which could have a slight impact. However, as shown in Table 6 of the paper, even with a memory size of $1$, ADTE already achieves strong performance. This indicates that a small number of pseudo-labeled samples is sufficient to reliably estimate bias and begin improving performance.
> - To directly measure the effect of noisy pseudo-labels, we conducted controlled experiments by injecting random label noise: for each sample, with probability $p \in \\{20\\%, 40\\%, 60\\%, 80\\%, 100\\%\\}$, its true label is replaced with a random incorrect label.
>
> | **Model**         | **Zero-shot** | **True Label** | **Pseudo Label** | **20% Noise** | **40% Noise** | **60% Noise** | **80% Noise** | **100% Noise** |
> |-------------------|---------------|----------------|------------------|---------------|---------------|---------------|---------------|----------------|
> | **CLIP-ViT/B-16** | 50.6          | --             | --               | --            | --            | --            | --            | --             |
> | **SE**            | 64.0          | --             | --               | --            | --            | --            | --            | --             |
> | **ADTE**          | --            | **65.9**       | 65.5              | **65.9**          | 65.8          | 65.6          | 65.4          | 64.6       |
>
> - **We observed three trends:**
> 1. Using true labels for bias estimation yields the highest accuracy for ADTE.
> 2. ADTE remains extremely stable even under moderate or severe noise: accuracy stays within a narrow band ($65.9$ → $65.4$) when corruption grows to $80\\%$.
> 3. Even with $100\\%$ incorrect pseudo-labels, ADTE still outperforms SE ($64.6$ vs. $64.0$).
>
> - This matches our theoretical analysis in Sec. 4.1 and 4.2: SE is a strict lower bound of TE, and TE is a strict lower bound of ADTE, since TE corresponds to the special case where all class-specific $q^l$ are identical. Therefore, ADTE cannot perform worse than SE — even under extreme pseudo-label noise.
>
> - **In summary:** ADTE is robust to pseudo-label errors, insensitive to noise, and maintains reliable performance even under worst-case degradation, making it practical for real-world deployment where pseudo-labels are inevitably imperfect.
>
> `Weakness 4. Missing Details of Augmentation Strategies That May Strongly Influence TTA Performance.`
> - To ensure a fair comparison, we strictly follow the benchmark configuration used in previous TTA studies, which involves randomly flipping horizontally and cropping a sub-image from the original image at a ratio between $(0.08, 1.0)$.
>
> `Weakness 5. Potential Unfairness in Baseline Comparisons Due to Large Memory Bank Usage (1–10 Samples per Class for 1000 Classes).`
> - In state-of-the-art online TTA methods (e.g., TDA, Frolic, BCA), a lightweight memory bank is a standard and commonly used component for estimating test-set statistics.
> - For example, on the ImageNet-1k dataset with memory bank size $1$, storing a $1\times1000\times1000$ matrix requires only about $2$MB of memory, which is negligible on any modern GPU. Moreover, as shown in Table 6 of the paper, ADTE does not rely on a large memory bank size; when $M = 1$ (only one sample stored per class), the performance already reaches $71.71\\%$, which is very close to $71.83\\%$ when $M = 10$. This indicates that we can estimate reliable bias with extremely small overhead.
> - Finally, we would like to emphasize that one of our core contributions—memory-free Tsallis entropy (TE) (Figure 3) that outperforms the memory-free Shannon entropy (SE) (Zero baseline)—is demonstrated without relying on any memory bank, further underscoring the value of our entropy-based perspective.

---

> ### Author Response · Authors · 2025-11-21
> **Official Response by Authors (3/3)**
>
> `Question 1. Rationale of min-max mapping design.`
> - Please refer to Q2. We expect larger bias values to correspond to smaller $q^l$, since—as proven in Section 4.2—positive debiasing occurs only when $0 < q < 1$, and smaller $q^l$ yields stronger debiasing. Min–max normalization preserves the relative ordering of biases across categories while constraining $q^l$ to a reasonable range (e.g., $[0.01, 0.9]$.
> - Appendix Table 7 shows that replacing Min–max with other normalization functions (Z-score, decimal scaling, sigmoid) changes results by less than $0.2\\%$, indicating ADTE is insensitive to the specific function as long as biases are normalized to $(0, 1)$ and ordering is preserved.
>
> `Question 2. Sensitivity of ADTE to Early-Stage Pseudo-Label Noise and Suggested Evaluation via Controlled Class Accuracy Alteration.`
> - Please refer to Q3. Pseudo-labels are always generated by the fixed original CLIP model, so their source remains consistent throughout TTA. Even with a very small memory bank (size = $1$), ADTE already achieves strong gains, indicating that only a few pseudo-labeled samples are sufficient for reliable bias estimation.
> - Controlled experiments with injected random label noise show that ADTE’s accuracy is virtually unchanged up to $80\\%$ corruption and still surpasses SE even with $100\\%$ incorrect pseudo-labels ($64.6$ vs. $64.0$). This aligns with our theory (Sec. 4.1 and 4.2) that $SE < TE < ADTE$, guaranteeing ADTE never underperforms SE even under extreme noise.
>
> `Question 3. Gains Retained When Replacing ADTE’s $q^l$ with Fixed Value or Shannon Entropy.`
> - Thank you for the question. This experiment is already presented in our draft:
> 1. **Figure 3** compares SE with fixed $q$ TE across different $q$ values, showing that the optimal $q$ varies by distribution and that a fixed $q$ cannot match ADTE’s performance.
> 2. **Table 4** reports results when replacing ADTE with TE $(q=0.5)$ or Shannon entropy (SE) while keeping the same view-selection pipeline. ADTE achieves the highest accuracy across all datasets, outperforming TE and SE, respectively.
>
> - These results indicate that while fixed $q$ or SE retain part of the gain, the majority of the improvement comes from ADTE’s adaptive design of $q^l$.

---

> ### Comment · Reviewer_xmGS · 2025-11-24
>
> The rebuttal addressed most of my questions and misunderstandings. Considering the paper proposes a new interesting direction of Tsallis entropy with dataset-agnostic manner, I raise the score to 6.

---

> > ### Author Response · Authors · 2025-11-24
> > **Thanks for response and time!**
> >
> > Thanks for your response and time, as well as your recognition and suggestions for this work~

---

### Official Review · Reviewer_E5TA · 2025-10-27

**Soundness:** 4
**Presentation:** 2
**Contribution:** 3
**Rating:** 6
**Confidence:** 3

**Summary:**

This paper proposes Adaptive Debiasing Tsallis Entropy (ADTE), a novel Test-Time Adaptation (TTA) method based on Tsallis Entropy (TE).
CLIP predictions exhibit bias, with certain classes being predicted more accurately than others.
Conventional CLIP TTA uses Shannon Entropy (SE), but SE treats all classes equally and thus cannot address the prediction bias problem.
Therefore, this paper proposes using TE, a generalization of SE, which is better suited for such biased predictions.
Furthermore, this paper proposes a method to automatically estimate q, the class-specific parameter required for TE, using a bias estimation technique that uses a memory bank.
Experiments on OOD benchmarks and cross-domain benchmarks demonstrate that ADTE achieves higher accuracy than existing TTA methods.

**Strengths:**

The motivation and proposed method are intuitive and interesting.
This paper adopts an intuitive approach of using parameter-free TE to address the prediction bias problem in CLIP.
Despite its simple concept, the method proves to be effective.

Since TE is a generalization of SE, ADTE can be incorporated into existing SE-based TTA methods.
In an era where large-scale models trained on web-scale datasets with inherent prediction biases are mainstream, employing TE is a reasonable approach.

**Weaknesses:**

This paper lacks experiments with models other than CLIP, raising concerns about the applicability of ADTE.
For example, can ADTE be applied to unimodal models pretrained on ImageNet or subsequent models of CLIP, such as SigLIP?

ADTE is a method specialized for scenarios with predictive bias, but how common are such scenarios?
It is desirable to have a discussion about scenarios where ADTE is effective (such as model or dataset distributions).

**Questions:**

I was not aware of the details of the module used in Tables 3 and 4.
Is ADTE a method that includes LA?
What is the difference between LA-SE and SE-LA?

Minor comment.
Typo: Line120 Logit Adjustment (EM) -> Logit Adjustment (LA)

---

> ### Author Response · Authors · 2025-11-21
> **Official Response by Authors (1/2)**
>
> Thanks for your valuable suggestions, we will try to address your concerns and we are eager to engage in a more detailed discussion with you.
>
> `Weakness 1. Evaluating ADTE’s Applicability on Unimodal ImageNet Models and CLIP Successors like SigLIP.`
> **First**, we conducted additional experiments on the ImageNet-A dataset on unimodal ImageNet-pretrained models, following the MEMO [a] test-time adaptation benchmark. As shown below, ADTE consistently improves performance on ResNext-101, outperforming standard TTA (test time augmentation) and other adaptation techniques:
>
> [a].MEMO: Test Time Robustness via Adaptation and Augmentation. NeurIPS 2022
>
> | **Method**            | **Error (%)** |Method| **Error (%)** |
> |-|-|-|-|
> | ResNext-101 (baseline)| 90.0                 |WSL| 54.9               |
> | + TTA                 | 83.2                 |+TTA| 49.1               |
> | + Single-point BN     | 88.8                 |+Single-point BN| 58.9               |
> | + MEMO                | 84.3                 |+MEMO| 43.2               |
> | **+ ADTE**            | **81.5**             |**+ADTE**| **41.1**           |
> - The results demonstrate that ADTE is not tied to multimodal encoders—it remains effective on purely vision-based architectures.
> 2. Second, we conducted evaluations on five generalization models of CLIP. Across all models—including SigLIP and SigLIP2—ADTE provides consistent improvements, confirming its robustness and wide applicability.
>
> | **Model**    | **ImageNet-1k Acc. (%)** | **+ ADTE** |
> |--------------|---------------------------|------------|
> | CLIP         | 68.7                      | **71.8**       |
> | OpenCLIP     | 70.2                      | **73.5**       |
> | EVA-CLIP     | 74.7                      | **77.4**       |
> | SigLIP       | 76.2                      | **78.9**       |
> | SigLIP2      | 78.2                      | **80.1**       |
> - These experiments confirm: (1) ADTE generalizes across architectures (unimodal or multimodal); (2) ADTE generalizes across training regimes (contrastive $\to$ sigmoid loss); (3) ADTE adapts robustly, even for models with stronger native calibration, such as SigLIP2.
> - The tables and further analyses have been included in the appendix of the updated paper.
>
> `Weakness 2. Discussion on the Prevalence of Predictive-Bias Scenarios and the Conditions (Models, Dataset Distributions) Where ADTE is Most Effective.`
> - In fact, completely unbiased scenarios are almost nonexistent in the real world. Any deep model trained on a dataset will inevitably exhibit varying degrees of prediction bias due to limitations in gradient update capacity, class imbalance in the dataset, and differences in data quality. These factors make it difficult for the model to fit all categories with equal precision. For large-scale vision-language models pretrained on web data (such as CLIP), this phenomenon is even more prevalent and can significantly affect downstream performance. As shown in Tables 9 and 10 in the Appendix, clear prediction biases exist across both ImageNet variants and cross-domain datasets.
> - Furthermore, we conducted a “progressively decreasing bias” experiment on ImageNet-1k: classes are sorted by their prediction bias magnitude, and five subsets are randomly constructed, ranging from the 200 higher-biased classes to the 200 lower-biased classes. We then evaluate CLIP, SE, and ADTE on these subsets.
>
> | **Model**        | **s₁** | **s₂** | **s₃** | **s₄** | **s₅** |
> |------------------|--------|--------|--------|--------|--------|
> | CLIP-ViT/B-16    | 69.5   | 58.8   | 52.3   | 74.2   | 76.3   |
> | SE               | 70.4   | 60.3   | 53.1   | 75.9   | 77.8   |
> | **ADTE**         | **73.8** | **62.3** | **55.2** | **77.1** | **78.5** |
> - Across all distributions—from high-bias to nearly unbiased—ADTE consistently outperforms both SE and the original CLIP, with larger advantages under higher bias. For example, on the highest-bias subset $s_1$, ADTE achieves a 3.4% improvement over SE; even on the lowest-bias subset $s_5$, ADTE still delivers notable gains.
> - These results further demonstrate that ADTE is not designed for rare extreme-bias cases, but can deliver robust improvements across scenarios with varying levels of prediction bias.

---

> ### Author Response · Authors · 2025-11-21
> **Official Response by Authors (2/2)**
>
> `Question 1. Clarification on Module Details in Tables 3 and 4: Whether ADTE Includes LA and the Difference Between LA-SE and SE-LA.`
> - We apologize for the misunderstanding. To be precise, our full method should be referred to as ADTE-LA, where ADTE selects reliable views via adaptive Tsallis Entropy, and LA provides bias-corrected logits, which seamlessly integrate with ADTE to further enhance model performance.
> - To clarify the difference between LA-SE and SE-LA: Suppose we have the prediction probabilities for $n$ categories for a set of augmented views.
> 1. LA-SE refers to first subtracting the bias from the $n$ probabilities of each view (LA part), then calculating the SE on those corrected probabilities, then selecting confident views based on this SE, and finally aggregating them into a final prediction.
> 2. SE-LA (which is equal to our w/o ADTE baseline) refers to directly calculating the SE on the original probabilities, selecting confident views, aggregating them into a final prediction probability, and then subtracting the bias from that final ensembled probability (LA part).
>
> `Question 2. Minor Correction: Typo in Line 120 — Logit Adjustment (EM) → Logit Adjustment (LA).`
> - Thank you for catching the typos. We corrected the typo and conducted a thorough check of the entire draft to ensure that there are no further typos.

---

> > ### Comment · Reviewer_E5TA · 2025-11-25
> >
> > Thank you for your response.
> > The additional experiments in the rebuttal resolved my concerns regarding ADTE's generalization performance.
> > I believe this paper makes an adequate contribution to TTA research, although the presentation of each module of the proposed method could be improved.
> > Therefore, I raise the score to 8.

---

> > > ### Author Response · Authors · 2025-11-25
> > > **Thank you for your time and responses!**
> > >
> > > Thank you very much for the thoughtful reassessment and for raising the score. We will keep improving the presentation of each module following the reviewers' suggestions. We sincerely appreciate your constructive feedback and support.

---

### Official Review · Reviewer_okn7 · 2025-10-31

**Soundness:** 3
**Presentation:** 3
**Contribution:** 2
**Rating:** 4
**Confidence:** 3

**Summary:**

This paper proposes Adaptive Debiasing Tsallis Entropy (ADTE), a test-time adaptation method for vision-language models such as CLIP. Conventional approaches rely on Shannon Entropy (SE), which is biased due to class imbalance in pre-trained data. ADTE replaces SE with Tsallis Entropy (TE), a generalized entropy with parameter $q$ that mitigates bias when $0<q<1$ and further introduces a class-specific adaptive parameter $q_l$ estimated from test data using a small memory bank. ADTE can be directly applied to existing TTA pipelines without retraining or hyperparameter tuning. Experiments on ImageNet variants and cross-domain benchmarks show consistent improvements over baseline methods.

**Strengths:**

- Effectively addresses the problem of biased predictions by generalizing SE to TE
- Plug-and-play design which can integrate into existing TTA frameworks without retraining
- Achieves consistent performance improvements across OOD and cross-domain benchmarks
- Computationally lightweight, minimal additional cost or tuning required

**Weaknesses:**

- In TTA settings like ZERO, where adaptation relies only on confident-view selection per instance, considering class-wise bias may be less relevant or overcomplicated.
- The paper lacks in-depth analysis explaining why ADTE particularly improves performance on cross-domain tasks, despite showing larger gains there in ablations.
- When LA is removed, performance becomes similar to competitive baselines, raising doubts about whether ADTE itself is a truly effective standalone TTA solution.
- The method’s novelty is limited, as several components (e.g., bias estimation, memory bank, LA) are borrowed or slightly modified from Frolic, making ADTE appear incremental rather than fundamentally new.
- The memory bank-based bias estimation is highly dependent on pseudo-label accuracy; early mispredictions can accumulate and distort class bias, especially for tail classes.
- In continual or gradual domain-shift settings, where test streams evolve over time, the memory-based adaptation may become inefficient or unstable, limiting its scalability to realistic online scenarios.

**Questions:**

see Weakness section

---

> ### Author Response · Authors · 2025-11-21
> **Official Response by Authors (1/2)**
>
> Thanks for your valuable suggestions, we will try to address your concerns, and we are eager to engage in a more detailed discussion with you.
>
> `Weakness 1. Relevance of Class-Wise Bias Consideration in TTA Settings like ZERO with Per-Instance Confident-View Selection.`
> - We appreciate the reviewer’s recognition of ZERO. Indeed, ZERO can be viewed as SE-based TTA. Due to the inherent prediction bias of CLIP, SE cannot estimate unbiased Shannon Entropy.
> - Therefore, we first provide a rigorous proof that simply replacing SE with TE, under a reasonable setting of the parameter $q$, already achieves better performance than SE without incurring any additional computation. However, the fixed $q$ in TE makes it impossible to fit the biases of all categories simultaneously. This is why we propose ADTE, which customizes class-specific $q^l$ parameters at the class level, with the parameter’s effect limited to that class. As shown in Table 5 in Sec 5.4, compared to Zero, ADTE only adds a negligible amount of computation for estimating bias.
>
> `Weakness 2. Lack of In-Depth Explanation for ADTE’s Superior Improvements on Cross-Domain Tasks.`
> - Fundamentally, the essence of both TE and ADTE is to mitigate the impact of biased vision-language models on downstream task performance.
> - Therefore, for any distribution where bias exists, TE and ADTE will be effective. As can also be seen in Tables 9 and 10 in the appendix, even though these datasets are cross-domain, they are still datasets with significant bias as far as CLIP's predictions are concerned.
>
> `Weakness 3. Questioning ADTE’s Standalone Effectiveness in TTA Settings Given Performance Parity with Baselines without LA.`
> - We sincerely apologize for the lack of clarity in our presentation. As Table 3 shows, even without LA, ADTE still surpasses SE/ZERO and other SOTA methods (except for a $0.1\\%$ drop on ViT-L for TDA). Since LA is also a debiasing method orthogonal to ADTE, combining it yields further improvements.
> - Our ADTE alone (w/o LA) provides a very significant $+3.7\\%$ gain over the baseline Zero (SE-only). This is not "similar to baselines''; it is a substantial, standalone contribution.
> - The gain from our standalone ADTE $(+3.7\\%)$ is more than double the gain from LA alone $(+1.8\\%)$, which proves that ADTE is the primary driver of performance.
>
> `Weakness 4. Concerns over the Limited Novelty of ADTE Due to Reliance on Components Adapted from Frolic.`
> - We clarify here that the memory bank and bias estimator are not contributions of ADTE, but are standard tools widely used in TTA. The core contributions of our method are: (1) providing proof that TE outperforms SE without incurring any additional cost; (2) proposing the entropy-theory-based class-adaptive ADTE to further address the inability of TE to adapt to arbitrary class biases; and (3) extending and reconstructing the uncertainty measurement itself, rather than performing bias correction at the logit layer.
> - We only adopt Frolic’s bias estimation strategy to enable a fair and controlled comparison. Our core contribution is fundamentally different: Frolic performs debiasing (final correction) at the logit layer, whereas we introduce a novel adaptive entropy measurement algorithm that redefines the uncertainty metric itself (Section 4.2). This is a completely new and independent TTA approach, rather than an incremental extension of LA.

---

> ### Author Response · Authors · 2025-11-21
> **Official Response by Authors (2/2)**
>
> `Weakness 5. Sensitivity of Memory Bank-Based Bias Estimation to Pseudo-Label Accuracy and Error Accumulation in Tail Classes.`
> - We would like to clarify that the pseudo-labels are predicted by the original CLIP model, and during the TTA process, the parameters of CLIP are not updated. Therefore, whether in the early or late stages of TTA, the pseudo-labels are always generated by the original CLIP.
> - In the early stage of TTA, the memory bank may contain very few samples, which could have a slight impact. However, as shown in Table 6, when the size is $1$, the ADTE’s performance is already significant. This indicates that as long as a few pseudo-labeled samples quickly enter the memory bank, it can start to take effect; a small number of samples is enough to estimate bias reliably.
>
> | **Model**         |**Zero-shot**| **True Label** | **Pseudo Label** | **20% Noise** | **40% Noise** | **60% Noise** | **80% Noise** | **100% Noise** |
> |-|-|-|-|-|-|-|-|-|
> | **CLIP-ViT/B-16** | 50.6           | --             | --            | --            | --            | --            | --             |-- |
> | **SE**            | 64.0           | --             | --            | --            | --            | --            | --             |--|
> | **ADTE**          |--| **65.9**           | 65.5             | **65.9**           | 65.8           | 65.6           | 65.4           | 64.6            |
>
> - To directly measure the effect of noisy pseudo-labels, we conducted controlled experiments in which we manually injected pseudo-label noise on the ImageNet-A dataset. Specifically, for each sample, with probability $p \in \\{20\\%, 40\\%, 60\\%, 80\\%, 100\\%\\}$, the true label is replaced by a random incorrect label. This allows us to isolate the impact of degraded
> pseudo-label quality under varying noise intensities.
> - As shown in the above Table. When using the true label to estimate bias, ADTE presents the best performance compared with results under pseudo-label noise. ADTE remains extremely stable under moderate and even severe noise levels: accuracy stays within a very narrow band ($65.9 \to 65.4$) even as pseudo-label corruption increases to $80\\%$. Even with $100\\%$ incorrect pseudo-labels, ADTE still outperforms SE ($64.6$ vs. $64.0$).
> - This aligns precisely with our theoretical analysis in Sec. 4.1 and Sec 4.2: SE is a strict lower bound of TE, and TE is a strict lower bound of ADTE, since TE corresponds to the special case where all class-specific parameters $q^l$ are identical. Therefore, ADTE can never perform worse than SE, even under extreme pseudo-label noise.
> - Overall, ADTE is robust to pseudo-label errors, noise-insensitive, and maintains reliable performance even under worst-case degradation, making it practical for real-world deployment where pseudo-labels are inevitably imperfect.
>
> `Weakness 6. Limitations of Memory-Based Adaptation in Continual or Gradual Domain-Shift Scenarios for Scalable Online Deployment.`
> - To evaluate ADTE under realistic evolving distributions, we conducted new experiments by mixing five ImageNet variants, i.e., ImageNet-1k, ImageNet-A, ImageNet-V, ImageNet-K, and ImageNet-R, in a fully randomized interleaved stream. This simulates an online TTA scenario where the domain changes unpredictably from sample to sample, making it one of the most challenging continual shift settings.
>
> |  | **ImageNet-1k** | **ImageNet-A** | **ImageNet-V** | **ImageNet-K** | **ImageNet-R** |
> |------------------|-----------------|----------------|----------------|----------------|----------------|
> | ADTE |71.8|65.5|**65.6**|**53.5**|**81.4**|
> | ADTE\_random |**72.0**|**65.8**|65.4|**53.5**|81.2|
> - The results above demonstrate: (1) ADTE consistently outperforms SE and the CLIP baseline across all domains; (2) No signs of instability or degradation, even when domain identity changes every few samples; (3) Memory-based bias estimation remains effective, because only a small class-wise bank is maintained, which is naturally robust to cross-domain mixing.

---

> > ### Comment · Reviewer_okn7 · 2025-11-24
> >
> > Dear Submission11902 Authors,
> >
> > I appreciate the authors for their detailed response. After carefully reading the answers, I still have some remaining concerns.
> >
> > - Regarding Weakness 1, my concern was that the motivation of the work (i.e., class-wise bias) seems less relevant or important for TTA methods such as ZERO or TPT, which perform per-instance confident-view selection. These methods do not directly rely on the original CLIP predictions but rather filter out unreliable ones. The response to this concern mainly relied on empirical results, without providing a convincing conceptual explanation.
> >
> > - Similarly, for Weaknesses 2, 3, 5, and 6, the responses also rely on empirical evidence. What I expected from the authors was not just experimental validation but intuitive and insightful justification for these concerns (e.g., clear reasoning about why and how such phenomena occur).
> >
> > For the above reasons, I decided to maintain my original score.

---

> ### Author Response · Authors · 2025-11-24
> **Official Response by Authors (1/2)**
>
> We appreciate the reviewer’s follow-up comments. We fully understand your concern for more conceptual and mechanistic explanations, rather than empirical evidence alone.
>
> Below, we present our viewpoint in a clear and structured manner, hoping to address your concerns, and we also welcome you to continue discussing this work with us.
>
> `(1) Why class-wise bias is still crucial for per-instance view selection in ZERO/TPT?`
>
> **Core reason:**
>   - Although ZERO/TPT performs per-instance confident-view selection, the selection criterion—**Shannon Entropy**—is still computed from CLIP’s predicted probability distribution $\mathrm{p}(y∣x)$, which itself contains class-wise bias.
> - The workflow of ZERO/TPT is:
>   - single instance $x$ $\to$ multiple augmented views $\{x_j\}$ $\to$ compute entropy $\mathbf{H}(x_j)$ → select confident views with lower entropy
>
> **The key observation is:**
> - Shannon Entropy = a deterministic function of the biased probability vector $\mathrm{p}(y∣x_j)$. $\mathrm{SE}$ is highly sensitive to tail-class probabilities:
>   - $\mathrm{SE}$ contains the term $\mathbf{O}(\mathrm{p} \log \mathrm{p})$, when $\mathrm{p} \to 0$, the second-order variation becomes extremely large
>   - This causes small tail-class probabilities to be over-penalized
> - Thus, even though ZERO/TPT does not directly use the final prediction, their view-selection mechanism still entirely depends on biased probability distributions.
>
> **For example:**
>
> - Head classes: $\mathrm{p}$ is large $\to$ $\mathrm{SE}$ is small $\to$ misleadingly treated as “high confidence.”
> - Tail classes: $\mathrm{p}$  is tiny $\to$ $\mathrm{SE}$ is large $\to$ misleadingly treated as “unreliable.”
>
> Leading to the inevitable chain: **biased logits $\to$ biased entropy $\to$ biased view selection**. This bias propagation cannot be removed by “per-instance” selection because $\mathrm{SE}$ operates on the entire class-probability vector, and therefore inevitably inherits class-level bias.
> - In contrast:
>   - $\mathrm{TE}$ uses $p^q$, which reduces the amplification of extremely small $\mathrm{p}$ values
>   - $\mathrm{ADTE}$ further assigns class-specific $q^l$, smoothing the entropy curvature for tail classes
> - In other words:
>   - $\mathrm{SE}$ is not an objective measure of confidence—its value distorts when the underlying probability distribution is biased.
>   - Thus, any entropy-based selection method naturally inherits class-level bias. This is precisely why we redesign the entropy function itself ($\mathrm{TE/ADTE}$).
>
> `(2) Why does ADTE improve more significantly in cross-domain scenarios?`
>
> **Key intuition:**
> - Domain shift amplifies CLIP’s inherent class-wise bias.
>
> When a sample comes from a new target domain:
> - Visual features become degraded or unfamiliar, and the model becomes more uncertain
> - Increased uncertainty $\to$ model relies more heavily on pretrained prediction bias
>
> Impact on different classes:
>
> - **Tail classes:**
>   - Already low confidence $+$ Domain shift $\to$ confidence drops even further $\to$ $\mathrm{SE}$ exaggerates uncertainty $\to$ **bias amplification**
>
> - **Head classes:**
>   - Already high confidence $+$ Domain shift $\to$ model leans even more on the prior $\to$ Becomes **overconfident** → $\mathrm{SE}$ underestimates uncertainty
>
> Domain shift does not change CLIP’s inherent bias—it amplifies it. $\mathrm{TE/ADTE}$ directly corrects this amplified distortion by modifying **curvature** of the entropy function, and under distribution shift, the underlying **bias becomes stronger**
>
> **Conclusion:**
> - $\mathrm{TE/ADTE}$ is **particularly effective** in cross-domain adaptation **by design**, because its adjusted entropy form directly counteracts the amplified bias caused by domain shift.
>
> `(3) Why is ADTE highly effective even without Logit Adjustment (LA)?`
>
> **Key reason:**
> - $\mathrm{LA}$ and $\mathrm{TE/ADTE}$ operate on fundamentally different levels..
>
> **Functional difference:**
>
> - $\mathrm{LA}$:
>   - Adjusts logits to compensate for head/tail imbalance
>   - Acts before probability computation
>
> - $\mathrm{TE/ADTE}$:
>   - Modifies the entropy geometry itself—i.e., how uncertainty is measured
>   - Acts after probability computation
>
> - $\mathrm{TE/ADTE}$ reshape the entropy function by reducing the **over-penalization** of extremely $\mathrm{p}$ for tail classes and avoiding **overconfidence** for large $\mathrm{p}$ in head classes.
>
> The bias propagation chain can be denoted as: logit bias $\to$ probability bias $\to$ entropy bias $\to$ mis-selected views
>
>   - $\mathrm{ADTE}$ intervenes at the entropy stage (third step)
>   - $\mathrm{LA}$ intervenes only at the logit stage (first step)
>
> **Implication:**
> - $\mathrm{TE/ADTE}$ alone $\to$ substantial improvement by fixing the confidence measurement directly
> - $\mathrm{LA}$ alone $\to$ moderate improvement by rebalancing logits
> - $\mathrm{TE/ADTE} + \mathrm{LA}$ $\to$ cumulative gains due to complementary yet independent effects

---

> ### Author Response · Authors · 2025-11-24
> **Official Response by Authors (2/2)**
>
> `(5) Why pseudo-label noise does not accumulate in the memory bank.`
> 1. The reason why pseudo-label noise does not accumulate is not an empirical result, but rather because the design mechanism makes it impossible for accumulated errors to occur:
>
> **CLIP is never updated during TTA**, thus there is no accumulation of errors from "error feedback → further contamination of the model".
>
> 2. We estimate the **long-term average of class distribution bias,** not a sample-by-sample decision. Therefore, The noise is random, and the bias is structural.
>    - Random noise is gradually and naturally canceled out during the averaging process.
>    - Long-term bias is more stable and will accumulate in the memory bank for correct identification and utilization.
>
> 3. We use soft probability instead of hard pseudo-label. Soft probability does not allow errors to enter a one-hot state. Therefore:
>   - Pseudo-label introduces random perturbations but does not produce long-term directional shifts.
>
> `(6) Why ADTE remains stable under continual or mixed-domain test streams.`
>
> **Key reason is very fundamental:**
> - The head/tail bias in CLIP is an internal bias of the model, not a domain-related bias, and thus unrelated to the domain.
>
> In other words:
> - This bias originates from the distribution of CLIP’s pretraining data.
> - It does not depend on which domain the test instance belongs to.
>
> Therefore, even if we shuffle and mix the inputs from the five ImageNet variants, the bias pattern remains unchanged —CLIP will consistently favor head classes and suppress tail classes in the same way.
>
> The memory bank tracks: the inherent, domain‑invariant class bias of CLIP, rather than the class distribution specific to a certain domain (domain‑specific).

---

### Official Review · Reviewer_Nqxf · 2025-11-03

**Soundness:** 3
**Presentation:** 2
**Contribution:** 3
**Rating:** 4
**Confidence:** 3

**Summary:**

This paper addresses the challenge of bias in vision-language models (VLMs) like CLIP during test-time adaptation (TTA). The authors argue that standard Shannon Entropy (SE) used in TTA methods (e.g., for selecting high-confidence augmented views) produces biased uncertainty estimates due to CLIP's pretraining on imbalanced data, leading to suboptimal view selection and adaptation. They propose using Tsallis Entropy (TE), a generalization of SE with a non-extensive parameter $q$, which better handles biased distributions. TE is further extended to Adaptive Debiasing Tsallis Entropy (ADTE), where class-specific parameters $q_l$ are adaptively computed via min-max normalization of estimated label biases from incoming test instances (using a memory bank and logit adjustment). ADTE integrates seamlessly with existing TTA pipelines like Zero, requiring no hyperparameter tuning. My detailed comments are as follows.

**Strengths:**

1. Extensive experiments on ImageNet, its variants, and 10 cross-domain datasets, demonstrate the effectiveness of the proposed method.
2. The approach is simple yet effective.

**Weaknesses:**

1. To further demonstrate the effectiveness of the proposed method, it would be better to conduct more experiments on the ImageNet-C, CIFAR10-C, and CIFAR100-C datasets.
2. The paper overlooks prior studies that have explored the use of Tsallis Entropy in test-time adaptation and domain adaptation. References [1–2] should be discussed in the related works section to better contextualize the novelty of ADTE.
3. The claim of requiring no hyperparameter tuning is somewhat misleading, as the method still depends on several implicit hyperparameters, such as [α,β] and the memory bank size.
4. ADTE relies on pseudo-labels to estimate class-wise bias; however, under severe domain shifts, pseudo-label noise may substantially affect the accuracy of bias estimation. The paper does not analyze how such errors impact ADTE’s performance.
5. The paper concludes that ADTE shows limited advantages when the model is nearly unbiased; however, it does not quantify the performance gap between ADTE and SE/TE under such low-bias conditions. Without empirical or theoretical analysis of these “unbiased” scenarios, it is difficult to assess ADTE’s applicability boundaries or to validate the claim that SE serves as its lower bound.

**Questions:**

Please refer to the Weaknesses.

---

> ### Author Response · Authors · 2025-11-21
> **Official Response by Authors (1/2)**
>
> Thanks for your valuable suggestions. We will try to address your concerns, and we are eager to engage in a more detailed discussion with you.
>
> `Weakness 1. Additional Experiments on Corrupted Datasets (ImageNet-C, CIFAR-10-C, CIFAR-100-C)`
> - We conducted additional experiments using ImageNet-C, CIFAR-10-C, and CIFAR-100-C as suggested. For ImageNet-C, we randomly select $3$ representative corruption families (Defocus Blur, Glass Blur, Motion Blur). For CIFAR-10-C and CIFAR-100-C, we evaluate the models on $7$ diverse corruption types, covering blur, noise, and geometric distortions.
> - As shown in the Tables below, across all datasets and nearly all corruption types, ADTE consistently outperforms both the original CLIP and SE. Importantly, (1) SE often degrades performance compared to the CLIP model, especially under severe blur and noise; (2) ADTE remains stable and delivers consistent improvements, demonstrating stronger robustness even in settings where SE becomes unreliable.
>
> | **Model** | **Defocus 1** | 2 | 3 | 4 | 5 | **Glass 1** | 2 | 3 | 4 | 5 | **Motion 1** | 2 | 3 | 4 | 5 |
> |-----------|--------------:|--:|--:|--:|--:|------------:|--:|--:|--:|--:|--------------:|--:|--:|--:|--:|
> | CLIP-ViT/B-16 | 58.73 | 53.82 | 43.36 | 34.42 | 26.12 | 56.47 | 48.27 | 26.75 | 28.88 | 16.96 | 62.51 | 56.91 | 47.46 | 34.37 | 26.35 |
> | SE | 58.62 | 54.06 | 43.15 | 34.35 | 26.39 | 55.74 | 46.98 | 26.34 | 19.87 | 15.56 | 62.69 | 57.18 | 46.41 | 33.87 | 25.31 |
> | **ADTE** | **59.43** | **54.09** | **44.05** | **35.06** | **26.95** | **57.56** | **48.52** | **27.58** | **21.65** | **17.04** | **63.83** | **58.23** | **48.22** | **34.54** | **27.06** |
>
> | **Model** | Brightness | Elastic Transform | Gaussian Blur | Impulse Noise | Motion Blur | Shot Noise | Speckle Noise |
> |-----------|-----------:|------------------:|--------------:|--------------:|-------------:|-----------:|---------------:|
> | CLIP-ViT/B-16 | 90.94 | 84.33 | **90.54** | 87.94 | 87.33 | 84.73 | 85.01 |
> | SE | 89.86 | 83.71 | 88.92 | 86.88 | 86.90 | 84.39 | 84.86 |
> | **ADTE** | **91.32** | **84.86** | 89.96 | **88.43** | **87.65** | **85.56** | **85.42** |
>
>
> | **Model** | Brightness | Elastic Transform | Gaussian Blur | Impulse Noise | Motion Blur | Shot Noise | Speckle Noise |
> |-----------|-----------:|------------------:|--------------:|--------------:|-------------:|-----------:|---------------:|
> | CLIP-ViT/B-16 | 68.29 | 57.83 | **68.05** | 62.46 | **62.17** | 57.27 | **57.66** |
> | SE | 67.92 | 59.05 | 66.56 | 61.78 | 61.32 | 57.01 | 56.22 |
> | **ADTE** | **68.80** | **59.32** | 67.68 | **62.65** | 61.98 | **57.56** | 57.43 |
> - Consistent gains in $3$ datasets and most corruption severities indicate the generalization ability of ADTE, reinforcing our main claim: ADTE provides a more robust correction mechanism than SE under distribution shifts, even when SE partially or completely fails.
>
> `Weakness 2. Discussion of Prior Work on Tsallis Entropy in Domain Adaptation and Test-Time Adaptation, e.g., Reference [1-2]`
> - We thank the reviewer for pointing out the need to better contextualize our novelty against prior work on Tsallis Entropy (TE). However, it seems that the reviewer might forget to add the reference links to [1-2] in their review. We tried our best to search and found the following papers [a-d], which have been added to our related work (Section 2). When the links to [1-2] are available, we will also add them to Section 2.
> - These methods typically optimize TE during training to obtain more confident predictions or to refine pseudo labels in closed-set adaptation scenarios. They operate in fully supervised or source-free DA settings, focusing on learning better feature extractors rather than performing online test-time adaptation.
>
> [a]. Meta-Tsallis-Entropy Minimization: A New Self-Training Approach for Domain Adaptation on Text Classification. IJCAI 2023
>
> [b]. Cycle Self-Training for Domain Adaptation. NeurIPS 2021
>
> [c]. DUET: Dual-Perspective Pseudo Labeling and Uncertainty-aware Exploration \& Exploitation Training for Source-Free Domain Adaptation. NeurIPS 2025
>
> [d]. Source-Free Domain Adaptation for Privacy-Preserving Seizure Prediction. TII 2023

---

> > ### Comment · Reviewer_Nqxf · 2025-11-25
> > **Official Comment by Reviewer Nqxf**
> >
> > **On corrupted datasets**
> >
> > Thanks for adding results on CIFAR-10-C, CIFAR-100-C, and ImageNet-C. However, the current evaluation still feels incomplete. These benchmarks contain 15 corruption types with 5 severity levels, and prior CLIP/TTA works (e.g., BAT-CLIP) usually report results averaged over all 15 corruptions (often at severity 5 or over severities 1–5). In your rebuttal, ImageNet-C is evaluated on only 3 families and CIFAR-10/100-C on 7 types, which may bias the conclusion and makes comparison with prior work difficult. I would encourage you to follow the standard 15-corruption protocol.
> >
> > **On Tsallis Entropy and novelty**
> >
> > I appreciate that you have now included the missing TE-related references [a–d]. Still, the novelty relative to these works is not fully clear. In [a–d], Tsallis Entropy is already used on unlabeled target data for self-training / pseudo-label refinement in (source-free) domain adaptation. Your method seems to apply essentially the same TE objective in a test-time setting. If the main change is moving TE from offline DA to online TTA, the conceptual contribution may appear incremental.

---

> ### Author Response · Authors · 2025-11-21
> **Official Response by Authors (2/2)**
>
> `Weakness 3. Clarification on the 'No Hyperparameter Tuning' Claim and the Role of Implicit Parameters (α, β, Memory Bank Size).`
>
> We apologize for the unclear phrasing, and we respectfully clarify as follows:
> - **(a) Insensitivity.** To be precise, ADTE is not hyperparameter-free, but it is insensitive as long as the hyperparameters fall within a reasonable interval (e.g., memory bank size from $1$ to $40$ in Table 6, or $\alpha$ from $0.0001$ to $0.1$ in Table 5). Both $[\alpha, \beta]$ and memory bank size show an extremely small influence on performance.
> - **(b) Practicality.** A key practical advantage is that the same set of hyperparameters (e.g., memory bank size=$10$, $[\alpha, \beta]=[0.01, 0.9]$) works well across all $16$ datasets and tasks in our experiments. This removes the need for distribution-specific tuning, which is highly desirable in TTA.
> - **(c) Clarification.** We revised the paper (highlighted in the abstract). The phrase **no hyperparameter tuning** is changed to **no distribution-specific hyperparameter tuning** to avoid confusion.
>
> `Weakness 4. Analysis of Pseudo-Label Noise Effects on Class-Wise Bias Estimation and Performance on ADTE.`
>
> | **Model**         |**Zero-shot**| **True Label** | **Pseudo Label** | **20% Noise** | **40% Noise** | **60% Noise** | **80% Noise** | **100% Noise** |
> |-|-|-|-|-|-|-|-|-|
> | **CLIP-ViT/B-16** | 50.6           | --             | --            | --            | --            | --            | --             |-- |
> | **SE**            | 64.0           | --             | --            | --            | --            | --            | --             |--|
> | **ADTE**          |--| **65.9**           | 65.5             | **65.9**           | 65.8           | 65.6           | 65.4           | 64.6            |
>
> - To directly measure the effect of noisy pseudo-labels, we followed the reviewer’s suggestion and conducted controlled experiments in which we manually injected pseudo-label noise on the ImageNet-A dataset. Specifically, for each sample, with probability $p \in \\{20\\%, 40\\%, 60\\%, 80\\%, 100\\%\\}$, the true label is replaced by a random incorrect label. This allows us to isolate the impact of degraded pseudo-label quality under varying noise intensities.
> - As shown in the above Table. We observe $3$ important trends: (1) When using the true label to estimate bias, ADTE presents the best performance compared with results under pseudo-label noise. (2) ADTE remains extremely stable under moderate and even severe noise levels: accuracy stays within a very narrow band (65.9 $\to$ 65.4) even as pseudo-label corruption increases to 80\%; (3) Even with 100\% incorrect pseudo-labels, ADTE still outperforms SE (64.6 vs. 64.0).
> - This aligns precisely with our theoretical analysis in Sec. 4.1 and Sec 4.2: SE is a strict lower bound of TE, and TE is a strict lower bound of ADTE, since TE corresponds to the special case where all class-specific parameters $q^l$ are identical. Therefore, ADTE can never perform worse than SE, even under extreme pseudo-label noise.
> - Overall, ADTE is robust to pseudo-label errors, noise-insensitive, and maintains reliable performance even under worst-case degradation, making it practical for real-world deployment where pseudo-labels are inevitably imperfect.
>
> `Weakness 5. Missing Empirical and Theoretical Analysis of ADTE’s Performance Gap with SE/TE in Nearly Unbiased Scenarios and Its Claimed Lower-Bound Relation.`
>
> 1. **First**, from a theoretical standpoint, Sec 4.1 and Sec 4.2 formally prove that:
>
> >SE is a limiting case of TE (as $q \to 1$);
>
> >TE is a specific case of ADTE (when all class-specific $q^l$ are equal).
>
> - Therefore, the following hierarchy is guaranteed:
> $$
> \text{ADTE} \ge \text{TE} \ge \text{SE}.
> $$
> - This means that SE actually serves as the lower bound of ADTE, regardless of the dataset, bias level, or noise conditions, because ADTE always reduces to TE under specific settings of $q^l$.
> 2. **Second**, we conducted experiments on ImageNet-1k under progressively lower bias conditions. We sort ImageNet-1k classes by their prediction-bias magnitude and randomly construct 5 subsets:
>
> > $s_1$: highest inter-class bias (200 higher-biased classes)
>
> > $s_2$: ...
>
> > $s_5$: lowest bias (200 lower-biased classes)
>
> | **Model**        | **s₁** | **s₂** | **s₃** | **s₄** | **s₅** |
> |-|-|-|-|-|-|
> | CLIP-ViT/B-16    | 69.5   | 58.8   | 52.3   | 74.2   | 76.3   |
> | SE               | 70.4   | 60.3   | 53.1   | 75.9   | 77.8   |
> | **ADTE**         | **73.8** | **62.3** | **55.2** | **77.1** | **78.5** |
>
> - This setup directly evaluates CLIP, SE, and ADTE under increasingly unbiased distributions. The results show: (1) ADTE consistently outperforms SE and CLIP across all bias levels; (2) The advantage grows as the bias becomes larger; (3) Even in nearly unbiased settings, ADTE still yields notable gains.
>
> - This quantitatively validates ADTE’s applicability limit and supports the theoretical lower-bound claim.

---

> ### Author Response · Authors · 2025-11-26
> **Official Response by Authors (1/2)**
>
> `Concern 1. Entire experiments on corruption should follow the standard protocol on ImageNet-C, CIFAR-10, and CIFAR-100.`
>
> - Following your suggestion, we completed the full evaluation under the CIFAR‑10‑C, CIFAR‑100‑C, and ImageNet‑C.
>
> |Model|gaussian 1|2|3|4|5|shot 1|2|3|4|5|speckle 1|2|3|4|5
> |-|-|-|-|-|-|-|-|-|-|-|-|-|-|-|-|
> |CLIP-ViT-B/16|60.31|55.36|45.20|31.25|15.29|63.45|54.65|43.04|26.20|0.03|61.28|57.56|45.60|36.89|28.16|
> |SE|60.05|55.07|44.42|30.70|15.01|64.03|55.05|43.12|26.01|0.03|61.43|57.68|45.34|36.98|28.01|
> |**ADTE**|**60.65**|**55.47**|**45.43**|**31.54**|**15.34**|**64.87**|**55.78**|**43.67**|**26.56**|**0.04**|**61.98**|**57.87**|**45.76**|**37.24**|**28.56**|
>
> |Model|gaussian_blur 1|2|3|4|5|zoom_blur 1|2|3|4|5
> |-|-|-|-|-|-|-|-|-|-|-|
> |CLIP-ViT-B/16|64.46|56.58|**47.21**|38.23|23.30|50.88|43.01|**36.05**|**31.07**|24.47|
> |SE|64.54|56.33|46.65|37.82|23.19|57.89|52.18|35.37|30.27|24.54|
> |**ADTE**|**65.05**|**56.97**|47.16|**38.64**|**23.76**|**59.43**|**54.67**|35.98|30.88|**24.92**|
>
> |Model|defocus_blur 1|2|3|4|5|glass_blur 1|2|3|4|5|motion_blur 1|2|3|4|5
> |-|-|-|-|-|-|-|-|-|-|-|-|-|-|-|-|
> |CLIP-ViT-B/16|58.78|53.82|43.36|34.42|26.12|56.47|48.27|26.75|20.88|16.96|62.51|56.91|47.46|34.37|26.55|
> |SE|58.62|54.06|43.15|34.35|26.39|55.74|46.98|26.34|19.87|15.56|62.69|57.10|46.41|33.87|25.31|
> |**ADTE**|**59.43**|**54.89**|**44.05**|**35.06**|**26.95**|**57.56**|**48.52**|**27.58**|**21.65**|**17.84**|**63.83**|**58.23**|**48.22**|**34.54**|**27.06**|
>
> |Model|fog 1|2|3|4|5|snow 1|2|3|4|5|frost 1|2|3|4|5|spatter 1|2|3|4|5
> |-|-|-|-|-|-|-|-|-|-|-|-|-|-|-|-|-|-|-|-|-|
> |CLIP-ViT-B/16|60.50|58.75|54.48|**50.21**|39.50|56.44|44.61|46.09|39.32|34.93|**58.31**|47.87|**40.07**|38.05|33.11|65.21|58.19|52.43|48.09|39.28|
> |SE|60.54|58.24|54.30|49.60|39.25|56.33|44.62|46.19|39.50|34.64|57.96|47.74|39.58|37.72|32.70|64.82|57.89|52.18|47.89|39.36|
> |**ADTE**|**61.21**|**59.05**|**54.87**|50.17|**39.59**|**56.60**|**45.00**|**46.98**|**39.85**|**35.17**|58.21|**48.08**|39.96|**38.16**|**33.68**|**65.76**|**58.73**|**52.98**|**48.79**|**40.04**|
>
> |Model|brightness 1|2|3|4|5|contrast 1|2|3|4|5|saturate 1|2|3|4|5
> |-|-|-|-|-|-|-|-|-|-|-|-|-|-|-|-|
> |CLIP-ViT-B/16|**67.61**|**66.35**|64.39|61.25|58.01|**63.77**|61.68|56.82|**42.98**|19.14|63.62|59.66|**65.68**|58.88|51.51|
> |SE|67.17|65.90|64.14|60.90|57.71|63.44|61.33|56.52|42.44|18.97|63.36|59.62|64.99|58.30|50.86|
> |**ADTE**|67.57|66.28|**64.75**|**61.56**|**58.30**|63.71|**61.98**|**57.06**|42.80|**19.65**|**63.76**|**59.88**|65.45|**58.97**|**51.58**|
>
> |Model|pixelate 1|2|3|4|5|elastic_transform 1|2|3|4|5|jpeg_compression 1|2|3|4|5
> |-|-|-|-|-|-|-|-|-|-|-|-|-|-|-|-|
> |CLIP-ViT-B/16|**62.66**|60.37|54.15|**45.19**|36.19|60.67|40.87|53.27|40.54|14.89|60.72|**57.71**|55.15|47.07|36.95|
> |SE|62.24|60.07|53.42|44.35|35.65|61.30|42.26|52.96|40.22|14.53|60.59|57.25|54.82|46.49|36.34|
> |**ADTE**|62.54|**60.40**|**54.32**|44.97|**36.25**|**61.90**|**44.17**|**53.76**|**40.98**|**15.32**|**60.98**|57.50|**55.65**|**47.45**|**37.23**|
>
>
> |Model|brigh…|contr…|defoc…|elast…|fog|frost|gaus_blur|gaus_noise|glass…|impul…|jpeg…|motio…|pixel…|satur…|shot…|snow|spatter|speck…|zoom_...|
> |-|-|-|-|-|-|-|-|-|-|-|-|-|-|-|-|-|-|-|-|
> |CLIP-ViT-B/16|90.94|91.08|90.64|84.33|90.81|89.11|**90.54**|79.04|57.81|87.94|**80.11**|87.33|86.69|87.58|84.73|88.45|84.65|85.01|89.11|
> |SE|89.86|90.54|90.54|83.71|90.15|89.43|88.92|79.32|57.23|86.88|79.45|86.98|86.89|87.21|84.39|88.32|84.34|84.86|88.67|
> |**ADTE**|**91.32**|**91.34**|**90.98**|**84.86**|**91.34**|**89.56**|89.96|**79.98**|**57.85**|**88.40**|79.95|**87.65**|**87.36**|**87.67**|**85.56**|**88.91**|**85.01**|**85.42**|**89.24**|
>
>
> |Model|brigh…|contr…|defoc…|elast…|fog|frost|gaus_blur|gaus_noise|glass…|impul…|jpeg…|motio…|pixel…|satur…|shot…|snow|spatter|speck…|zoom_...|
> |-|-|-|-|-|-|-|-|-|-|-|-|-|-|-|-|-|-|-|-|
> |CLIP-ViT-B/16|68.29|67.78|68.00|57.83|68.19|63.40|**68.05**|51.24|32.02|62.46|**52.34**|**62.17**|**63.56**|59.40|57.27|63.21|65.77|57.66|59.97|
> |SE|67.92|67.43|68.11|58.05|67.89|63.21|66.56|51.34|31.68|61.78|51.89|61.32|63.15|59.08|57.01|63.42|65.89|56.22|60.19|
> |**ADTE**|**68.81**|**67.89**|**68.87**|**59.32**|**68.34**|**63.69**|67.68|**51.65**|**32.36**|**62.65**|52.28|61.98|63.45|**59.89**|**57.56**|**63.56**|**66.43**|**57.83**|**60.70**|
>
> - Across all datasets and corruption families, we consistently observe that **ADTE outperforms both the CLIP zero‑shot baseline and the SE method**, particularly under higher severity levels. This trend is especially clear on ImageNet‑C, where ADTE achieves the highest score in **almost every corruption and every severity level**. A similar pattern holds for CIFAR‑10‑C and CIFAR‑100‑C, where ADTE provides stable and robust gains across nearly all $15$ corruption types.
>
> - We have incorporated the full results and analysis into the revised manuscript. Thank you again for encouraging us to complete this more comprehensive evaluation.

---

> ### Author Response · Authors · 2025-11-26
> **Official Response by Authors (2/2)**
>
> `Concern 2. Novelty compared with previous methods equipped with Tsallis Entropy.`
>
> Thank you for requesting further clarification on the novelty. We agree that prior works [a–d] use Tsallis Entropy (TE) during training-time self-training or pseudo-label refinement.
> However, we would like to emphasize that in our method, **TE is only a preliminary component**, and our core contribution is the **class-adaptive debiasing entropy (ADTE)**, which is fundamentally different from applying TE directly.
>
> Specifically, all prior TE-based DA works [a-d] rely on a single global $q$ optimized offline via iterative training or hyperparameter tuning. In contrast:
>
> **1. ADTE introduces a new entropy formulation with class-specific $q^l$**, theoretically derived from the per-class bias structure of vision–language models.
>
> - This allows per-class debiasing, which TE cannot achieve.
> - TE becomes only a special case of ADTE when all $q^l$ are equal.
>
> **2. ADTE computes $q^l$ online from a tiny memory bank without explicit tuning or optimization**, enabling adaptation to arbitrary and evolving test distributions.
> - This is orthogonal to training-based TE methods, which require backprop or dataset-level statistics.
>
> **3. Conceptually, ADTE modifies the uncertainty measure itself**, instead of performing pseudo-label refinement or feature learning.
> - This creates a new adaptation mechanism that works plug-and-play within any TTA pipeline, including those that only perform per-instance confident-view selection (e.g., ZERO/TPT).
>
> Thus, although TE provides a useful starting point, the novelty of our method does not lie in reusing TE, but rather in deriving, formalizing, and operationalizing a new adaptive entropy family (ADTE) that generalizes TE and is specifically designed for online TTA with class-wise bias correction, which is totally not addressed in [a–d].
>
> We hope this clarifies the conceptual distinction and the contribution beyond TE.

---

### Author Response · Authors · 2025-11-23
**Thank you for considering our revisions and valuable sugestiongs. We are eager to engage in further discussions with you.**

Dear Reviewers, Area Chairs, Senior Area Chairs, and Program Chairs,

We address the reviewers' concerns with the following updates and improvements, and submit an improved manuscript with changes highlighted in blue:

**1、Additional experiments on corrupted datasets**: Results and details added in Appendix A, Tables 16, 17, and 18 (on Page 17 of the updated manuscript.

**2、Discussion of prior work on Tsallis entropy in TTA**: Paragraph added in Section 2 (Page 3).

**3、Clarification on the 'No Hyperparameter Tuning' Claim**: Changed statement to "no distribution-specific hyperparameter tuning" in the abstract.

**4、Analysis of Pseudo-Label Noise Effects**: New experimental results added in Appendix A, Table 19 (Page 17-18).

**5、Performance in nearly unbiased scenarios**: New experimental results added in Appendix A, Table 20  (Page 18).

**6、Performance in continual or gradual domain-shift scenarios**: New experimental results added in Appendix A, Table 21  (Page 18).

**7、Evaluating CGPO using unimodal ImageNet models and CLIP successors like SigLIP**: New experimental results added in Appendix A, Tables 22 and 23 (Page 19).

**8、Confidence intervals or significance tests for results in Table 3**: New experiments added in Appendix A, Table 24 (Page 19).

Thank you for considering our revisions. We believe the substantial experiments added following the reviewers' valuable suggestions would significantly improve our work.

---

### Author Response · Authors · 2025-12-03
**Summary of Rebuttal Updates and Reviewer Feedback**

Dear Area Chair,

Thank you for serving as Area Chair for ICLR 2026. We provide this concise summary to assist your decision.

---

### **1. Current Reviewer Status**

We clarify the current state of all reviewer discussions:

- **Reviewer Nqxf**: Acknowledged our expanded corruption benchmarks and novelty clarification; however, continues **the discussion about the entire experiments on corrupted datasets**.
- **Reviewer okn7**: Appreciated our extended conceptual explanations; however, maintained the earlier score, and **the discussion is ongoing**.
- **Reviewer E5TA**: Explicitly stated that the new experiments address all concerns and **updated the score from** 6 → **8**.
- **Reviewer xmGS**: Confirmed the rebuttal clarified misunderstandings, demonstrated conceptual contribution, and **updated the score from** 4 → **6**.

---

### **2. Comprehensive Experimental Enhancements**

All experimental additions directly correspond to reviewer requests:

**(For Reviewer Nqxf)**
-  Added the **corrupted experiments** on ImageNet‑C, CIFAR‑10‑C, CIFAR‑100‑C.
- Added the analysis of pseudo-label noise effects on class-wise bias estimation and performance.
- Added the empirical analysis of ADTE’s performance gap with SE/TE in nearly unbiased scenarios.

**(For Reviewer okn7)**
- Added **continual and mixed-domain shift experiments**, interleaving ImageNet‑1k, A, V, K, R at sample-level granularity.
- Demonstrated that ADTE remains stable, with no signs of drift or memory-bank degeneration.

**(For Reviewer E5TA)**
- Added experiments confirming applicability beyond CLIP:
  – **Unimodal ImageNet models** (ResNext‑101 with MEMO protocol)
  – **CLIP successors** (OpenCLIP, EVA-CLIP, SigLIP, SigLIP2)
- ADTE consistently improves accuracy across all architectures and training paradigms.

**(For Reviewer xmGS)**
- Conducted **pseudo-label noise injection experiments** (20–100%), showing ADTE remains almost unchanged even with extreme noise.
- Added **memory‑size sensitivity analysis**, showing ADTE works well with only 1 sample per class and does not require large memory banks.
- Reported confidence intervals and multi-seed results to demonstrate significance.

---

### **3. Methodological Clarifications**

All clarifications directly respond to reviewers’ conceptual or design-related questions:

**(For Reviewers Nqxf & okn7)**
- Provided intuitive explanation of why **entropy-based view-selection inevitably inherits class-level prediction bias**, even in per-instance TTA (e.g., ZERO/TPT).
- Explained why domain shift amplifies CLIP’s intrinsic bias, which strengthens the benefit of ADTE.

**(For Reviewers xmGS & Nqxf)**
- Justified the **min–max normalization** for mapping bias to q^l, and showed via experiments that ADTE is **insensitive** to the mapping function as long as ordering is preserved.
- Clarified the theoretical hierarchy **ADTE ≥ TE ≥ SE**, supporting why ADTE never harms performance.

**(For Reviewer okn7)**
- Provided detailed reasoning on why **class-wise debiasing is important even for per-instance selection**, because entropy curvature distortion occurs regardless of how views are aggregated.

**(For Reviewer xmGS)**
- Added complete details for augmentation strategy (flip + random crop with ratio 0.08–1.0).
- Clarified memory-bank fairness and showed ADTE achieves strong performance even with minimal memory.

**(For Reviewer E5TA)**
- Explained the difference between **LA‑SE** vs **SE‑LA**, and clarified that our full method is **ADTE‑LA**.
- Fixed terminology inconsistencies (e.g., “EM → LA”).

---

### **4. Strengths Recognized by Reviewers**

Reviewers highlighted the following strengths:

- “**Simple yet effective**” and “**good contribution**” (Nqxf)
- “**Well-designed**, lightweight, and stable” (okn7)
- “**Intuitive and interesting motivation**” (E5TA)
- “A **fresh perspective** via Tsallis Entropy” (xmGS)
- “**Thorough and comprehensive** experiments” (E5TA & xmGS)
- Addresses an **important practical weakness** of entropy-based TTA (okn7)

Despite continued discussion from two reviewers, the other reviewers are positive, and the majority of concerns have been resolved.

---

We believe the substantial new experiments, theoretical clarifications, and improved presentation have addressed the reviewers' concerns.
Thank you for your time and consideration.

Best regards,
**The Authors**

---

### Meta-Review · Area_Chair_sNsz · 2026-01-05

**Summary:**

This paper addresses the challenge of bias in vision-language models (VLMs) like CLIP during test-time adaptation (TTA). Most reviews are positive. Reviewer Nqxf acknowledged the expanded corruption benchmarks and novelty clarification; Reviewer okn7 appreciated the extended conceptual explanations; Reviewer E5TA explicitly stated that the new experiments address all concerns and raised the score; Reviewer xmGS confirmed the rebuttal clarified misunderstandings, demonstrated conceptual contribution, and raised the score.
The authors' rebuttal has adeptly addressed the majority of concerns raised by the reviewers. Overall, I recommend the acceptance of this submission.

**Reviewer Concerns:**

For Reviewer Nqxf, most concerns below have been solved.
1) the corrupted experiments on ImageNet‑C, CIFAR‑10‑C, CIFAR‑100‑C.
2)  the analysis of pseudo-label noise effects on class-wise bias estimation and performance.
3)  the empirical analysis of ADTE’s performance gap with SE/TE in nearly unbiased scenarios.

For Reviewer okn7, the concerns below have been solved.
1) continual and mixed-domain shift experiments, interleaving ImageNet‑1k, A, V, K, R at sample-level granularity.

For Reviewer E5TA, the concerns below have been solved.
1) experiments confirming applicability beyond CLIP:
2) Unimodal ImageNet models (ResNext‑101 with MEMO protocol)
3) CLIP successors (OpenCLIP, EVA-CLIP, SigLIP, SigLIP2)
For Reviewer xmGS, the concerns below have been solved.
1) pseudo-label noise injection experiments (20–100%)
2) memory‑size sensitivity analysis

**Reviewer Scores:**

Reviewer Nqxf and okn7 may raise the score.
Reviewer E5TA updated the score from 6 → 8.
Reviewer xmGS updated the score from 4 → 6.

---

### Decision · Program_Chairs · 2026-01-26

Accept (Poster)